# MathlibLemma: Folklore Lemma Generation and Benchmark for Formal Mathematics

**Xinyu Liu** [1]   **Zixuan Xie** [1]   **Amir Moeini** [1]   **Claire Chen** [2]   **Shuze Daniel Liu** [3 4]   **Yu Meng** [1]   **Aidong Zhang** [1]
**Shangtong Zhang** [1]

## Abstract

While the ecosystem of Lean and Mathlib has enjoyed celebrated success in formal mathematical reasoning with the help of large language models (LLMs), the absence of many folklore lemmas in Mathlib remains a persistent barrier that limits Lean's usability as an everyday tool for mathematicians like LaTeX or Maple. To address this, we introduce MATHLIBLEMMA, a modular LLM-based pipeline for automated folklore-lemma mining: the discovery, formalization, and proving of reusable intermediate facts that mathematicians often take for granted but that are not always present in formal libraries. At its core, MATHLIBLEMMA proactively mines the missing connective tissue of mathematics. The pipeline produces a verified library of folklore-style lemmas, including 1,506 Lean-checked proofs that pass a proof-bypass screen; a small curated pilot subset has also been merged into Mathlib, providing external evidence that selected outputs can meet expert library standards. Leveraging this pipeline, we further construct the MATHLIBLEMMA benchmark, a suite of 4,028 non-trivial type-checked Lean statements spanning a broad range of mathematical domains. By transforming the role of LLMs from passive consumers to active contributors, this work takes a step toward AI-assisted expansion of formal mathematical libraries.

## 1. Introduction

Fully formal proofs checked by proof assistants against a small trusted kernel offer a principled route to making mathematical verification reliable, scalable, and reproducible. This promise becomes increasingly salient as proofs in modern mathematics and computer science grow longer and more intricate. A striking illustration is the ABC conjecture: formulated independently by Masser (1985) and Oesterlé (1988), it is widely viewed as a master key in number theory. Despite its deceptively simple statement, a proof would reshape the landscape of number theory by unifying numerous disparate results, streamlining long and intricate arguments, and clarifying what is fundamentally possible and impossible. Mochizuki later announced a highly technical proof spanning hundreds of pages, as presented in (Mochizuki, 2021a;b;c;d). Yet, despite substantial effort by experts in the past 10 years, the community still has not reached a simple shared understanding of the argument's correctness (Tao, 2012; Scholze & Stix, 2018; Mochizuki, 2019; Rittberg, 2021). This situation is not merely sociological trivia: it exposes a structural limit of an informal pipeline that relies on human intuition, local norms, and bounded attention. Formal verification is designed precisely to turn such debates from sociological uncertainty to automatic verification.

At the same time, many researchers remain reluctant to write proofs in formal languages. Beyond the well-known high barrier to entry, a particularly stubborn bottleneck is library coverage: even when a proof is conceptually straightforward, formalizing it can stall on the absence of small "obvious" facts. In this paper we focus on Lean 4 (de Moura & Ullrich, 2021) and its flagship library Mathlib (The mathlib Community, 2020), which has become the de facto standard ecosystem for Lean formalization. Despite Mathlib's scale, users still encounter missing lemmas at exactly the wrong moment: in the middle of a proof whose mathematical ideas are clear but whose formal details require a long chain of routine rewrites, inequalities, or structural facts. We refer to these missing pieces as *folklore lemmas*: facts that working mathematicians routinely use without comment, often learned implicitly from textbooks, lectures, or repeated practice, but which are not necessarily present in the library in

---

[1]Department of Computer Science, University of Virginia [2]The Division of Physics, Mathematics and Astronomy, California Institute of Technology [3]Massachusetts Institute of Technology [4]Purdue University. Correspondence to:
Xinyu Liu <xinyuliu@virginia.edu>,
Shangtong Zhang <shangtong@virginia.edu>.

*Proceedings of the 43rd International Conference on Machine Learning*, Seoul, South Korea. PMLR 306, 2026. Copyright 2026 by the author(s).

a readily reusable form. The resulting friction constitutes a last-mile barrier that limits proof assistants from becoming everyday tools, comparable in convenience to LaTeX or Maple.

This last-mile phenomenon is widely reported by practitioners. For example, in Terence Tao's Lean formalization project on the polynomial Freiman–Ruzsa conjecture (Tao, 2023), he notes that the mathematically interesting parts were comparatively easy to formalize, while "technical obvious steps" took the longest, especially in the final stretch involving properties of independent random variables. Similarly, in a Lean formalization effort on foundational reinforcement learning theory, a single line about conditional expectation expands into a large formal development (Zhang, 2026), illustrating that folklore steps may be simple to state informally yet costly to reconstruct in a proof assistant. These last-mile gaps can also degrade LLM-based formal reasoning. When a necessary lemma is missing, an LLM cannot simply invoke it via a concise reference. Instead, it is forced to reconstruct the result from scratch, transforming a single inference step into a lengthy derivation. This can expand the search space and consume context tokens, making the overall proof attempt more prone to hallucination and failure.

In parallel to these challenges, automated theorem proving in Lean has advanced rapidly with large language models (LLMs) (Achim et al., 2025; Chen et al., 2025; Chervonyi et al., 2025; Lin et al., 2025; Math-Inc, 2025; Hubert et al., 2026; Lin et al., 2026). However, many existing evaluations of LLMs' capability in formal reasoning emphasize fixed problem sets (Zheng et al., 2022; Azerbayev et al., 2023; Tsoukalas et al., 2024; Yu et al., 2025) and are only a partial proxy for addressing the last-mile gap. Furthermore, those systems and evaluations largely consume Mathlib but do not systematically contribute to Mathlib. This one-way pattern leaves an important opportunity unexplored: using models not only to consume formal libraries, but also to help expand them.

We argue that addressing the last-mile gap calls for a complementary direction: rather than waiting for users to encounter gaps, we should proactively discover and formalize these folklore lemmas at scale. To that end, we introduce MATHLI-BLEMMA, **to our knowledge, the first LLM-based modular pipeline designed specifically for automated folklore-lemma mining from an existing formal library**.

At a high level, MATHLIBLEMMA consists of four LLM-backed modules. The first is a *Discovery Module* that identifies candidate missing folklore lemmas using existing files from Mathlib as seeds. This *Discovery Module* outputs candidate lemmas in Lean directly. The second is a *Judge Module* using LLM-as-a-judge to filter out Lean statements that are mathematically wrong. The third is a *Formalizer*

*Module* that tries to fix syntax and type errors of the Lean statements by interacting with a Lean server. All retained Lean statements after the *Formalizer* are type-checked. The last is a *Prover Module* that tries to prove the Lean statement that passes the judge and type-checks after formalization. This modular design is essential for the novel task of folklore mining: by decoupling semantic plausibility from syntactic correctness and proof search, it makes the main failure modes observable and separately addressable.

This framework constitutes our primary contribution, pioneering the task of automated folklore mining. Its efficacy is demonstrated by the production of a type-checked library of folklore lemmas, a subset of which **has already been formally merged into Mathlib**, providing external evidence that selected outputs can meet expert library standards.

As a second artifact, we construct the MATHLIBLEMMA benchmark, a suite of 4,028 non-trivial type-checked statements. To assess statement quality, we audit a seed-stratified sample of unproven residuals across diverse topics; **78%** of the audited instances are mathematically sound, suggesting that many benchmark instances left unsolved by current provers are valid rather than hallucinated. We evaluated state-of-the-art models including GPT-5.1, GPT-5.1 with low reasoning (OpenAI, 2025a;b)[1], Goedel-Prover-V2-32B (Lin et al., 2026), DeepSeek-R1-Distill-Qwen-32B, DeepSeek-R1-Distill-Llama-70B (Guo et al., 2025), Kimina-Prover-72B (Wang et al., 2025), and Qwen3-235B-A22B-Thinking-2507 (Yang et al., 2025). Collectively, 37% of the 4,028 lemmas are proved, whereas each individual model can prove about 19% at most.

These artifacts highlight the key distinction of our work: MATHLIBLEMMA is both a benchmark and a library-expansion pipeline. It requires limited manual annotation, can be refreshed across domains, and allows future prover successes to be recycled into a verified folklore-lemma repository rather than remaining only benchmark scores.

**Code and Data Availability.** The source code and datasets are available at `https://github.com/Sequentia l-Intelligence-Lab/MathlibLemma`. The repository contains two primary artifacts: the complete benchmark suite of 4,028 non-trivial type-checked statements, and the verified library of 1,506 Lean-checked generated proofs that pass the proof-bypass screen.

---

[1]OpenAI's GPT-5.1 API documentation lists four reasoning-effort settings: *none*, *low*, *medium*, and *high*. In this work, "GPT" refers to the *none* setting (which is the default configuration), and "GPT-Reasoning" refers to the *low* setting. We excluded *medium* and *high* levels as they were prohibitively slow and expensive for our large-scale evaluation. We note that previous baselines, such as Goedel or Kimina, typically did not include evaluations against these frontier reasoning APIs.

## 2. Related Work

Our work sits at the intersection of automated theorem proving, library evolution, and large language model (LLM) benchmarking. Unlike systems designed to solve individual Olympiad-level problems, MATHLIBLEMMA focuses on the structural last-mile challenge: constructing the dense lattice of folklore lemmas required for everyday formalization.

### 2.1. Lemma Synthesis and Library Expansion

The goal of automatically expanding formal libraries predates the recent progress in LLMs. Sivaraman et al. (2022) introduce data-driven methods to synthesize lemmas from interactive proof traces in Coq. With the rise of LLMs, focus shifted to generative conjecturing to grow libraries of reusable skills (Wang et al., 2024). More recent works employ LLMs to generate conjectures from existing library seeds. LeanConjecturer (Onda et al., 2025) uses rule-based context extraction and automated tactics (e.g., aesop, a powerful proof search tool) to filter and evaluate candidates in Lean. In parallel, Lemmanaid (Alhessi et al., 2025) targets Isabelle/HOL, combining LLM-generated templates with a symbolic engine to enforce well-typedness and novelty relative to the Archive of Formal Proofs (AFP).

MATHLIBLEMMA differentiates itself from recent library-centric works like LeanConjecturer by prioritizing *folklore mining*—the identification of missing structural gaps—over the general *conjecturing* of plausible facts. Rather than merely expanding the library with random valid statements, our Discovery Module targets the missing connective tissue that working mathematicians implicitly rely on but is absent from the library.

### 2.2. Intermediate Lemmas and Feedback-Driven Repair

A parallel line of research uses lemma generation as a mechanism for proof search, evolving rapidly from reinforcement learning to agentic repair loops. Early LLM-based approaches like ProD-RL (Dong et al., 2024) use reinforcement learning to reward the hierarchical decomposition of proofs. More recently, the state-of-the-art shifted to agentic loops that actively exploit compiler feedback. Systems like Delta Prover (Zhou et al., 2025) and APOLLO (Ospanov et al., 2025) utilize Lean's compiler error messages to iteratively refine subgoals. Meanwhile, ProofAug (Liu et al., 2025) analyzes proof structures in Isabelle, extracting and refining "maximal compatible semi-proofs" to improve robustness. Similarly, Seed-Prover (Chen et al., 2025) and Hilbert (Varambally et al., 2026) utilize recursive decomposition to generate intermediate scaffolding. Gödel's Poetry (Davis, 2025) then extends this approach by integrating AST-based parsing to programmatically extract and recursively prove subgoals.

MATHLIBLEMMA aligns with this feedback-rich paradigm but differs fundamentally in **artifact reusability**. Systems like Delta Prover or Seed-Prover typically generate lemmas as ad-hoc scaffolding customized to resolve specific hard problems, often discarding them or leaving them dependent on the problem context. In contrast, our pipeline is designed to produce reusable, general-purpose statements derived from library gaps. Our outputs are structurally designed not as transient steps in a proof search tree, but as permanent candidates for the library.

### 2.3. Benchmarks and the Saturation of Olympiad Math

Finally, our work addresses a critical gap in evaluation. The landscape of formal reasoning benchmarks has historically focused on Olympiad-level problem solving or undergraduate autoformalization (Zheng et al., 2022; Azerbayev et al., 2023; Tsoukalas et al., 2024; Achim et al., 2025). However, performance on these benchmarks is reaching saturation. Recent empirical advancements demonstrate that success rates on MiniF2F have surged from 50% (Wang et al., 2024) to near-saturation levels of 96% (Zhou et al., 2025). This rapid progress suggests that "depth" (solving hard problems) has reached a high level of proficiency, while "breadth" (knowledge coverage) remains unaddressed. MATHLIBLEMMA fills this void. Instead of testing ingenuity on solved benchmarks, we evaluate a component's ability to formalize and prove the vast quantity of routine background facts that human mathematicians take for granted. This shift from isolated problem solving to *foundational library building* is essential for the next stage of formal reasoning, ensuring that models can not only solve puzzles but also contribute productively to formal ecosystems. A recent work Xie et al. (2026) also follows this direction by benchmarking whether LLMs are able to judge whether a pull-request to Mathlib is ready to be merged.

## 3. The MATHLIBLEMMA Framework

### 3.1. Task Formulation: Automated Folklore Mining

We formalize the task of *Automated Folklore Mining*. Unlike standard autoformalization (which translates informal text to formal code) or Olympiad-level proving (which solves closed problems), folklore mining aims to **proactively expand the library** by identifying and verifying missing intermediate results.

Formally, let $\mathcal{L}$ denote a background formal library (e.g., Mathlib). Given a seed context $\mathcal{C} \subset \mathcal{L}$ (represented by a source file and its imports), our objective is to synthesize a set of pairs $\{(S, P)\}$, where $S$ is a lemma statement and $P$ is its proof, satisfying three conditions:

**1. Semantic Novelty:** The lemma $S$ represents a "missing step" or generalization not explicitly present in $\mathcal{L}$, avoiding

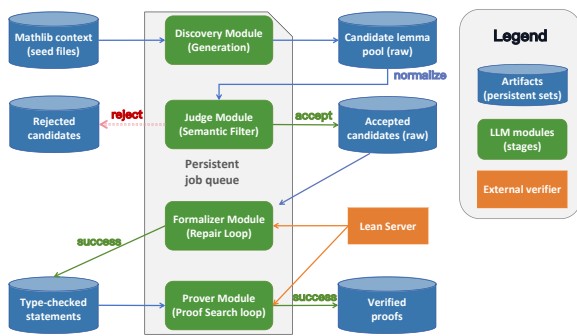

*Figure 1.* Overview of MATHLIBLEMMA. A modular pipeline where the Discovery Module mines candidates from Mathlib seeds, followed by semantic filtering (Judge), syntactic repair (Formalizer), and proof generation (Prover), yielding a verified library and benchmark.

trivial restatements of existing facts.

**2. Mathematical Plausibility:** The statement $S$ corresponds to a true mathematical fact, free from hallucinations or false conjectures, ensuring it is worth attempting to prove.

**3. Syntactic Validity:** The final output must be rigorously verified by the kernel, i.e., the statement $S$ is syntactically well-formed in context $\mathcal{C}$, and the proof $P$ is valid for $S$ under this context. For proof artifacts used in the benchmark and release, we additionally require the generated code to pass the proof-bypass screen defined below.

This formulation shifts the challenge from purely *solving* a given statement to *discovering* what statements are worth solving.

### 3.2. High-Level Pipeline

Figure 1 provides a visual overview of this architecture. The detailed execution logic is provided in Algorithm 1 (Appendix A.1). In this section, we detail the specific goal and method for each module. To ensure reproducibility, we document the exact prompt templates, input specifications, and structured output extraction logic in Appendix B, with detailed runtime configurations (e.g., temperature, max tokens) provided in Appendix A.2.

**Design Rationale: Factorizing Failure Modes.** Standard end-to-end synthesis entails a compounding risk of failure, where mathematical hallucinations, syntax errors, and proof search failures are entangled. Our four-stage modular pipeline is designed to **factorize these distinct failure modes** into orthogonal stages: (1) The **Judge Module** eliminates semantic noise early, preventing the prover from wasting compute on mathematically false premises; (2) The **Formalizer Module** bridges the *syntax-semantics mismatch*, strictly separating the **syntactic formalization of the statement** from the logical burden of proving it; (3)

Consequently, the **Prover Module** operates on a **guaranteed well-formed proposition**, focusing solely on proof construction without being blocked by invalid definitions or imports. This separation allows for specialized feedback loops that are impossible in monolithic generation.

#### 3.2.1. DISCOVERY MODULE: CONTEXT-AWARE CANDIDATE GENERATION

**Goal.** Generate diverse candidate lemma statements conditioned on Mathlib seed context $\mathcal{C}$, without attempting proofs. Each candidate is emitted as Lean code containing lemma/theorem declarations with `sorry` (a placeholder for missing proofs).

**Method.** Given the full content of a seed Mathlib file, the Discovery Module employs a structured prompt to brainstorm plausible missing folklore lemmas. This prompt is explicitly designed to encourage the **identification of structural gaps and cross-topic interactions** (e.g., via high-level planning) before generating the formal statements. Crucially, to ensure engineering robustness, the module enforces a normalized output format, which facilitates the reliable extraction and splitting of diverse candidates for downstream verification.

#### 3.2.2. JUDGE MODULE: SEMANTIC FILTERING (LLM-AS-A-JUDGE)

**Goal.** Filter out statements that are mathematically false or nonsensical *before* invoking kernel-validated checking or proof search.

**Method.** For each single-lemma candidate, we ask an LLM to assess mathematical correctness while explicitly ignoring Lean well-formedness. This design ensures that valid mathematical insights are not prematurely discarded due to trivial, fixable syntax errors, which are effectively handled by the subsequent **Formalizer Module**. The **Judge Module** outputs a binary verdict (`correct` vs. `wrong`); we discard `wrong` candidates and forward only `correct` ones, providing a lightweight semantic filter before kernel-level repair or proof search.

#### 3.2.3. FORMALIZER MODULE: KERNEL-GUIDED TYPE CHECKING AND REPAIR

**Goal.** Transform a semantically plausible candidate into a well-formed Lean statement $S$ that is a valid proposition in context $C$ (still with `sorry`).

**Method.** For each judged lemma, we invoke kernel checking via Kimina Lean server (Dos Santos et al., 2025) to elaborate and type-check the statement and return structured diagnostics. If errors occur, we iteratively prompt an LLM to fix them using a transcript that includes (i) the current Lean code and (ii) diagnostic messages. The prompt forbids

proving the lemma and focuses solely on making the statement compile (e.g., resolving namespaces, identifiers, and implicit arguments). The loop terminates when the Lean server reports no error-level diagnostics or when a maximum number of repair trials is reached. Successful results are stored together with generation metadata (e.g., trial counts) to facilitate downstream benchmarking and analysis.

### 3.2.4. PROVER MODULE: CLOSING THE LOOP WITH VERIFIED PROOFS

**Goal.** Produce a valid proof term $P$ for the type-checked statement $S$ such that the resulting artifact passes Lean checking and the proof-bypass screen.

**Method.** Given a type-checked lemma statement, we prompt a prover LLM to (i) propose a proof plan and (ii) output the full lemma-with-proof in Lean using a stage-specific marker. We validate each attempt by compiling the generated lemma-with-proof via the Lean server. A proof attempt is accepted only if it satisfies both Lean checking and a proof-bypass screen. First, the self-contained Lean file must elaborate with no error-level diagnostics. Second, the screen is applied to the extracted model-generated proof code, not to the imported Mathlib environment. After stripping Lean comments, it rejects incomplete-proof placeholders (`sorry`, `admit`, or Lean's corresponding `sorryAx`) and model-introduced top-level declarations that could add new assumptions, primitive objects, or black-box facts (`axiom`, `constant`, or `opaque`). Fully proved auxiliary lemmas are allowed, but the model may not expand the trusted context with new assumptions or hidden declarations. On failure, we capture the **compiler error messages** returned by the server. We then append the **failed code snippet combined with these error messages** to the history buffer and prompt the model to repair the proof. This generate-verify-repair loop starts with an initial proof attempt and then allows up to two repair rounds; we report this setting as Success@2. Successful proofs are stored persistently for downstream evaluation.

## 4. The MATHLIBLEMMA Benchmark

Using the pipeline described in Section 3, we constructed MATHLIBLEMMA, a benchmark of 4,028 non-trivial type-checked Lean 4 statements derived from diverse domains of Mathlib. Unlike previous datasets focused on high-school competitions (e.g., MiniF2F) or undergraduate text problems (e.g., ProofNet), our benchmark focuses on the *missing intermediate steps*—the folklore lemmas—that bridge high-level intuition and formal verification.

### 4.1. Evaluation Protocol

While our system's goal is *discovery* (Section 3.1), the benchmark evaluates a model's capability for *closed-book automated proving* on these discovered lemmas. Each benchmark instance is a pair $(\mathcal{C}, S)$, consisting of a **well-typed Lean proposition** $S$ together with the **sufficient self-contained context** $\mathcal{C}$ required to elaborate it (including necessary imports, open namespaces, and variable declarations). Importantly, we do *not* provide ground-truth proofs; the task is to synthesize a proof $P$ strictly from the given statement and context.

**Input and Output.** The input provided to the model includes both the context $\mathcal{C}$ and the statement $S$. Explicitly providing $\mathcal{C}$ ensures that the instance is **self-contained** and compilable without requiring the model to retrieve external files. Given this input, a model must output Lean code that proves the statement. A prediction is counted as **successful** only if the generated proof passes both Lean checking and the proof-bypass screen defined in Section 3.2.4.

**Closed-book Constraint.** During evaluation, models are not allowed to use external retrieval tools (e.g., code search, RAG, or browsing Mathlib sources). This design serves three key purposes: 1. **Isolating Model Capability**: By removing the dependency on external retrievers, we decouple the model's reasoning power from the quality of the search stack, ensuring a fair "apples-to-apples" comparison across diverse architectures. 2. **Feasibility**: It allows for scalable evaluation of expensive frontier reasoning models (e.g., GPT-Reasoning), where multi-turn agentic retrieval for thousands of instances would be cost-prohibitive. 3. **Conservative Lower Bound**: This setting acts as a stress test for the models' internalized mastery of Mathlib. While it conflates reasoning with recall, the resulting scores represent a conservative lower bound on performance, measuring robust formalization ability rather than search efficiency.

### 4.2. Construction and Filtering

We construct the MATHLIBLEMMA benchmark by executing the first three stages of our pipeline (Discovery, Judge, Formalizer) on seed contexts sampled from Mathlib. Specifically, we selected a diverse set of 109 source files spanning the domains shown in Figure 2. For each instance, the entire source file is used as the seed context. To ensure the high quality of the benchmark statements, we employ a frontier model (**GPT-5.1**) for all generation and filtering steps. This process functions as a rigorous **automated funnel**: the system generates raw candidates, semantically filters hallucinations via the Judge Module, and syntactically repairs them via the Formalizer Module. Only candidates that survive this pipeline are admitted into the benchmark, ensuring high fidelity without the bottleneck of manual annotation.

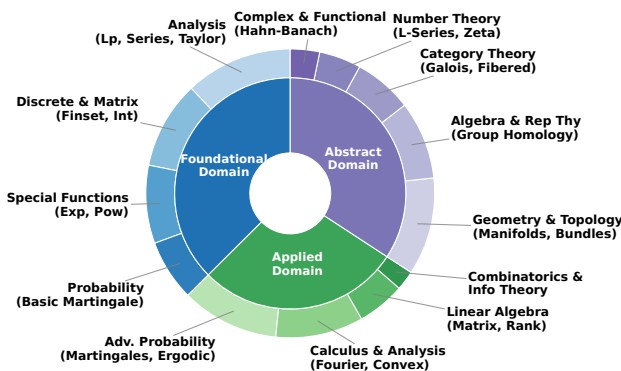

*Figure 2.* MATHLIBLEMMA taxonomy of seed-topic areas. The benchmark is partitioned into three distinct domains (inner ring): **Foundational**, **Applied**, and **Abstract**. The outer ring shows topic areas used to source seed contexts, with representative examples in parentheses.

**Data Stratification.** To facilitate fine-grained analysis, we stratified the seed contexts into three distinct domains: Foundational, Applied, and Abstract. This classification was performed manually by grouping Mathlib source files according to their primary mathematical subject matter and directory location.

**1. Foundational Domain.** This domain covers the "bread and butter" of formalized mathematics: real analysis, discrete structures (e.g., sets, matrices), basic probability, and others. These lemmas often require high familiarity with Mathlib's naming conventions and standard tactics, testing a model's *internalized knowledge recall* capabilities.

**2. Applied Domain.** This domain includes domains such as advanced probability (martingales), information theory, and convex analysis. While the mathematical concepts are often intuitive to humans (e.g., "expected value is linear"), they are notoriously difficult to formalize due to heavy type-class constraints (e.g., measurability proofs, integrability conditions). This domain tests a model's ability to handle syntactic overhead and formalization gaps.

**3. Abstract Domain.** This domain spans abstract fields including category theory, algebraic topology, and differential geometry. Proofs here rely less on calculation and more on abstract structural reasoning (e.g., diagram chasing, functoriality). This tests a model's capacity for deep logical reasoning.

We use this partition to report construction statistics and evaluate model performance.

**Pipeline Funnel Statistics.** Table 1 reports the detailed statistics of this funnel across the three domains. Observe that the drop from 'Proposed' to 'Judge-accepted' is significant (filtering out 30-37% of raw candidates), highlighting the critical role of the Judge Module in removing halluci-

nations. To assess this filter, we audited a seed-stratified sample of 53 candidates rejected by the Judge. We found that 54% were indeed mathematically false, while the remaining 46% were valid statements erroneously rejected (false negatives). Nevertheless, every seed context still contributed surviving candidates after judging. While this indicates an imperfection in the Judge's recall, it shows that the Judge acts conservatively, discarding many questionable candidates before formalization. The quality of the retained benchmark statements is assessed separately by the residual provability audit in Section 4.3. Subsequently, the Formalizer Module successfully compiles the majority (61%–77%) of these judged candidates. Notably, the compilation rate is highest in Foundational Domain (77.1%) and lowest in Abstract Domain (61.3%), reflecting the higher syntactic complexity of abstract mathematical structures.

*Table 1.* Pipeline Funnel Statistics. "Judge-accepted" denotes statements labeled correct by the Judge Module; "Compilable" denotes statements that passed the Lean kernel check. Percentages represent the pass rate relative to the count of the preceding stage (e.g., Compilable % is calculated relative to the Judge-accepted count).

| Domain | Proposed | Judge-accepted (%) | Compilable (%) | Trivial (%) |
|---|---|---|---|---|
| Foundational | 3427 | 2447 (71.4%) | 1887 (77.1%) | 4.1% |
| Applied | 2591 | 1726 (66.6%) | 1219 (70.6%) | 6.6% |
| Abstract | 3130 | 1977 (63.2%) | 1211 (61.3%) | 10.8% |

**Lightweight Triviality Check.** As a lightweight proxy for triviality and obvious redundancy, we flag instances solved immediately by `aesop` as trivial. The low percentage of trivial lemmas across domains (4.1% in Foundational Domain, increasing to 10.8% in Abstract Domain) indicates that the Discovery Module successfully targets non-trivial gaps rather than trivial tautologies. Unless stated otherwise, we exclude these trivial instances from the main proving results. In total, this removes 289 aesop-trivial statements (78 Foundational, 80 Applied, and 131 Abstract), leaving 4,028 non-trivial type-checked statements for the main benchmark.

**Generation-model dependence.** Because GPT-5.1 is used for candidate discovery, judging, and statement repair, the benchmark distribution partly reflects GPT-5.1's view of what folklore lemmas look like and how they should be phrased. The subsequent proof evaluation is objective in the sense that success is determined by Lean checking together with the proof-bypass screen, but the set of evaluated statements is shaped by the generation model. We therefore view MATHLIBLEMMA as a versioned benchmark generated by a specified pipeline, rather than as a model-independent census of all missing Mathlib folklore. The current Discovery stage is intentionally seed-file local and one-shot, without repository-wide retrieval, dependency-graph traversal, or proof-attempt-driven gap detection; preliminary open-model discovery runs produced lower-quality candidates, so

we leave broader repository-aware discovery to future work.

### 4.3. Validity: Provability Audit

Unlike benchmarks derived from existing libraries, MATHLIBLEMMA consists of **novelly discovered conjectures**. Consequently, these instances do not inherently possess ground-truth proofs. An evaluation failure can therefore have two different causes: the statement may be valid but beyond the prover's current capabilities, or it may be unprovable as stated because of missing hypotheses or an overstrong conclusion. To separate these factors, we conducted an expert provability audit on up to two model-unsolved instances from each audited seed context, a seed-stratified sampling strategy that gives coverage across diverse Mathlib contexts. We use this audit to assess the validity of the sampled residuals and to diagnose common failure modes.

**Audit Protocol.** The audit was performed by expert formalizers. Unlike the closed-book evaluation for models, the human experts operated in an open-book setting, allowed to utilize the full Mathlib library, AI assistance, and external search tools. For each sampled instance, the expert attempted to construct a proof from scratch to determine its mathematical truth and formal provability. This setup ensures that we strictly measure the *intrinsic validity* of the statements, independent of the closed-book constraints imposed on the models.

**Audit Results.** Table 2 presents the results of this audit. Across all domains, experts successfully proved 78% of the sampled instances. This high validity rate primarily serves as evidence of our framework's efficacy, suggesting that the construction pipeline has reasonably high precision on this sampled subset and produces mathematically sound targets. Since the audit sample is drawn from model-unsolved residuals, the human success rate shows that many residual failures are not false statements; it highlights a gap between current closed-book provers and expert open-book formalization.

For the unproven minority (22%), qualitative analysis (Appendix D) reveals that the dominant failure mode is **Missing Hypotheses** (e.g., omitting integrability or non-emptiness conditions), rather than fundamental mathematical errors. This finding underscores a characteristic challenge of folklore mining: while models effectively capture the core mathematical *intuitions*, they occasionally overlook the implicit *boundary conditions* that human mathematicians take for granted but formal systems strictly require.

## 5. Experiments

We evaluate state-of-the-art LLM provers on MATHLIBLEMMA benchmark to measure their ability to bridge the last-mile formalization gap. Unlike static datasets (e.g.,

*Table 2.* **Human Provability Audit.** We analyze a seed-stratified sample of *model-unsolved* instances to assess residual validity across diverse seed contexts. For each domain, we sampled lemmas from diverse seed contexts (2 lemmas per seed). The high human success rate implies that the benchmark remains challenging but solvable, and that many failures are due to current prover limitations rather than false statements.

| Domain | Sampled | Human-proved | Rate |
|---|---|---|---|
| Foundational | 38 | 30 | 79% |
| Applied | 56 | 40 | 71% |
| Abstract | 44 | 37 | 84% |
| Total | 138 | 107 | 78% |

MiniF2F) where ground-truth proofs are known, our benchmark represents a **dynamic discovery setting**: instances are type-checked conjectures without pre-existing solutions. Consequently, success is defined strictly by **independent Lean checking together with the proof-bypass screen**. In this section, we primarily report the **Sequential Success@2** (specifically **Success@2**: cumulative success within a budget of 2 repair turns) as the definitive measure of system efficacy. Unlike standard Pass@$k$ which aggregates parallelizable independent samples, Success@$k$ measures sequential self-correction. Since this repair loop is inherently serial, we limit the budget to $k = 2$. Further experimental details and performance analyses are provided in Appendix C.

### 5.1. Experimental Setup

We evaluate a diverse set of models: the generalist GPT-5.1 ("GPT") and its reasoning variant ("GPT-Reasoning"); the open-weight reasoning models DeepSeek-R1-Distill-Qwen ("DS-32B"), DeepSeek-R1-Distill-Llama-70B ("DS-70B") (collectively "DeepSeek"), and Qwen3-235B-A22B-Thinking-2507 ("Qwen"); the library specialist Goedel-Prover-V2 ("Goedel"), a model pre-fine-tuned on Mathlib; and Kimina-Prover-72B ("Kimina"), a model trained via RL on formal competitions (e.g., MiniF2F). We follow the closed-book protocol defined in Section 4.1.

### 5.2. Main Results

We summarize the aggregated performance in Table 3.

**1. The "Specialist vs. Generalist" Trade-off.** The most striking result is the performance dichotomy between open-weight specialists and frontier generalists. **Goedel-Prover** achieves a remarkable **27.42%** success rate in the Foundational Domain, significantly outperforming GPT-Reasoning (22.44%). However, its performance collapses in the Abstract Domain, dropping to 8.24% (roughly half of GPT-Reasoning's 18.61%). Qualitative inspection suggests distinct failure modes: we found that **GPT's** primary weakness is **hallucination**—it frequently attempts to invoke non-

*Table 3.* **Main Results (Success@2).** Comparison of cumulative success rates (%) across the three domains. While **GPT-Reasoning** performs consistently well, specialized models like **Goedel** show strong performance on foundational tasks but degrade significantly on abstract math.

| Model | Foundational | Applied | Abstract | Total |
|---|---|---|---|---|
| GPT | 18.91 | 12.03 | 14.17 | 15.69 |
| GPT-Reasoning | 22.44 | **15.28** | **18.61** | **19.39** |
| Kimina | 13.93 | 7.64 | 8.52 | 10.70 |
| Goedel | **27.42** | 11.24 | 8.24 | 17.70 |
| DeepSeek-32B | 11.00 | 3.60 | 3.80 | 6.98 |
| DeepSeek-70B | 10.89 | 3.60 | 2.87 | 6.68 |
| Qwen | 1.93 | 2.63 | 3.15 | 2.46 |
| Union (All Models) | 47.48 | 27.48 | 30.93 | 37.39 |

existent APIs or lemmas, reflecting a lack of precise library knowledge. In contrast, **Goedel**, having been fine-tuned on Mathlib, rarely hallucinates syntax but suffers from **logical failures** in abstract settings, unable to construct the deep structural arguments required when "memorized" patterns do not apply. **DeepSeek**, despite its strong reputation in informal math reasoning, achieves only 6.98% success. This suggests that general mathematical reasoning ability does not automatically transfer to formal proving without explicit training on the target library's syntax and conventions (as seen in Goedel). Meanwhile, **Kimina** (10.70%), trained on competition problems (MiniF2F), underperforms Goedel (17.70%) on our folklore task. This suggests that **library-centric fine-tuning** (Goedel) is more effective for folklore mining than **competition-centric RL** (Kimina), as folklore lemmas resemble standard library code more than tricky olympiad puzzles.

**2. Universal Benefit of Test-Time Compute.** Test-time compute yields universal gains. Comparing GPT and GPT-Reasoning, we observe a consistent performance gain of 3.3–4.4 percentage points across domains. Unlike specialized models which exhibit high variance, this uniform improvement suggests that reasoning budget effectively facilitates self-correction regardless of domain specificity.

**3. Efficiency of Reasoning Distillation.** Comparing open-weight models reveals that *scale is not all you need*. DeepSeek-32B (6.98%) substantially outperforms Qwen (2.46%), while remaining close to the 70B variant (6.68%). This suggests that reasoning distillation and Lean-specific proof behavior may matter more than parameter scale alone in this setting.

**4. Diversity is More Valuable than Scale.** Perhaps the most critical observation is the massive gap between the best individual model (19.39%) and the Union performance (37.39%). The fact that the union coverage is **nearly twice** that of the strongest single model implies that these models are **highly orthogonal**: Goedel solves problems that GPT misses, and vice versa. This diversity gain strongly suggests

that future formal reasoning systems should prioritize **ensembling diverse architectures** (e.g., mixing specialized small models with reasoning-heavy frontier models) rather than relying on a single monolithic prover.

**5. The Remaining Folklore Gap.** The union of all models solves 37.39%, while applying the 78% human success rate from our seed-stratified residual audit suggests an indicative solvable fraction near 86%. Although this extrapolation is diagnostic rather than a precise benchmark-level estimate, the gap shows that MATHLIBLEMMA remains far from saturated and continues to expose limitations of current neural provers.

## 6. Practical Impact: The Verified Library

Beyond benchmarking model capabilities, MATHLIBLEMMA fulfills a more fundamental scientific goal: **the automated expansion of formal knowledge**. Our pipeline produces a large-scale repository of Lean-checked proofs that pass the proof-bypass screen, which constitutes a durable and reusable asset for the community. This verified library serves two distinct practical functions: first, it acts as an immediately usable **reference database** for human formalizers, offering valid proofs for over a thousand missing folklore facts; second, it functions as a **high-quality staging area** for Mathlib, supplying candidates that can be selectively curated for upstreaming when they are non-redundant, stated at the right level of generality, named consistently with the surrounding API, placed appropriately, and written in a maintainable proof style.

### 6.1. The Proved Subset

Using the Prover module (Section 3.2.4), we successfully synthesized 1,506 verified proofs from 4,028 non-trivial type-checked statements. A released proof is counted only if the self-contained Lean file elaborates without error-level diagnostics and the generated proof code passes our proof-bypass screen. This screen strips Lean comments and rejects two classes of bypass artifacts: incomplete-proof placeholders (`sorry`, `admit`, or Lean's corresponding `sorryAx`) and model-introduced top-level declarations that could supply new assumptions or black-box facts (`axiom`, `constant`, or `opaque`). Thus the proof must be constructed from the supplied Mathlib context and Lean-checked generated proof terms, rather than from incomplete placeholders, newly introduced assumptions, or black-box declarations.

We release these proofs as a comprehensive repository of machine-checked folklore, organized by domain and topic. By covering a diverse range of folklore-style statements, this repository can serve as a searchable auxiliary layer for downstream projects.

## 6.2. Upstreaming to Mathlib

To evaluate whether the generated lemmas can meet the standards of a real formal-mathematics library, we conducted a small pilot upstreaming effort to Mathlib. We selected six generated, Lean-verified lemmas across several areas of analysis, measure theory, and probability, manually cleaned their names and proof style, and submitted them as Mathlib pull requests. At the time of writing, three of these six lemmas have been merged into Mathlib, one was rejected as redundant with existing library material, and two were not pursued further within the paper timeline.

We emphasize that this pilot was intentionally small. Although our pipeline produced 1,506 Lean-checked proofs that pass the proof-bypass screen, submitting all of them directly to Mathlib would not be an appropriate integration strategy. Mathlib is maintained by a volunteer community, and bulk submission of AI-generated lemmas would impose substantial review burden without prior coordination. Moreover, Lean checking and proof-bypass screening establish formal validity of the generated proof artifacts in a fixed Lean environment, but Mathlib upstreaming additionally requires curation: a lemma should be non-redundant, stated at the right level of generality, named consistently with the surrounding API, placed in the appropriate file, and written in an acceptable proof style. We therefore use selective upstreaming as a high-standard validation signal rather than as a bulk-ingestion mechanism.

The three merged examples illustrate the kind of reusable folklore facts surfaced by the pipeline:

**Analysis.** A monotonicity lemma for the Grönwall bound, stating that the Grönwall time-bound function is monotone in time under nonnegativity assumptions on its parameters. Such facts are routinely used when propagating differential-inequality bounds.

**Measure theory.** A compatibility lemma for kernels, stating that restricting a constant kernel to a measurable set agrees with the constant kernel associated with the restricted measure. This is a small but useful API lemma connecting kernel restriction and measure restriction.

**Probability.** An almost-everywhere congruence lemma for central moments, stating that two real-valued random variables that are equal almost everywhere have the same central moment with respect to the same measure and moment order. This captures the standard principle that such probabilistic quantities depend only on the almost-everywhere equivalence class of the random variable.

These examples are given in Appendix E. A controlled retrieval-augmented downstream ablation remains a valuable next step. The present pilot serves a different purpose: it shows that selected generated lemmas can survive Mathlib curation and therefore are plausible components of the auxiliary folklore layer such an ablation would use.

## 6.3. Cost and Practicality

We also tracked approximate paid API cost. Discovery, Judge, and Formalizer cost about $200 for 4,028 non-trivial type-checked statements, or roughly $0.05 per statement. Proving cost about $500; after the proof-bypass screen, it yielded 1,506 released Lean proofs, or roughly $0.33 per proof for proving alone and $0.46 when amortizing the full $700 pipeline cost. These figures exclude author curation, local serving costs for open-weight models, and Mathlib maintainer review. Because the upstreaming pilot deliberately submitted only six curated lemmas, the three merged PRs are a high-standard validation signal rather than a denominator for cost-per-merged-lemma.

## 7. Conclusion

We introduced MATHLIBLEMMA, to our knowledge, the first LLM-based modular pipeline capable of proactively discovering and formalizing mathematical folklore. By systematically mining the "connective tissue" of mathematics, our work establishes a constructive methodology toward addressing the last-mile gap in formal libraries. The efficacy of this framework is validated by two critical pieces of evidence: first, the successful selective upstreaming of generated proofs into Mathlib provides external evidence that selected outputs can meet expert library standards; second, a rigorous human audit reveals that 78% of a seed-stratified sample of model-unsolved residuals are nonetheless provable, providing evidence that many generated model-unsolved challenges are mathematically sound. Complementing the framework, we released the MATHLIBLEMMA benchmark, which quantifies the distinct gap between human mathematical intuition and current formal reasoning capabilities.

We acknowledge four limitations and corresponding directions for future work. First, our novelty filters rely mainly on syntactic uniqueness and lightweight automation, so semantically equivalent restatements of existing Mathlib facts may remain; stronger checks should combine library retrieval, semantic search, and proof-based equivalence tests. Second, kernel elaboration verifies that a repaired statement is well formed, but not that it preserves the original mathematical intent; future repair loops could add semantic-consistency checks, in the spirit of autoformalization alignment methods (Gao et al., 2025). Third, MATHLIBLEMMA is tied to a fixed Lean/Mathlib snapshot and should be periodically rechecked as APIs evolve, with additional normalization for implicit type, universe, and typeclass choices. Finally, Lean-checked proofs still require curation for naming, placement, generality, documentation, and proof style.

## Impact Statement

This paper presents work whose goal is to advance the field of Machine Learning. A potential risk is that automatically generated lemmas could impose review burden or introduce misleading artifacts if used without curation. We mitigate this by requiring Lean kernel checking, a proof-bypass screen, and selective human review before upstreaming. We do not advocate bulk submission of generated lemmas to Mathlib without coordination with maintainers.

## Acknowledgements

This work is supported in part by the US National Science Foundation under the awards III-2128019, SLES-2331904, and CAREER-2442098, the Commonwealth Cyber Initiative's Central Virginia Node under the award VV-1Q26-001, a Cisco Faculty Research Award, and an Nvidia academic grant program award.

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

---

**Algorithm 1** Four-Stage Modular Lemma Generation Pipeline

---

**Input** : Set of seed contexts $\mathcal{T}$ (Mathlib files)
**Output**: Set of verified pairs $\{(S, P)\}$

```
// Stage 1:  Discovery (Context-Aware Generation)
```
**foreach** *context* $\mathcal{C} \in \mathcal{T}$ **do**
   $R \leftarrow \text{LLM}(\mathcal{C})$ ;                                `// Brainstorm raw candidates`
   Split $R$ into individual raw statements $\{r_1, r_2, \ldots, r_n\}$ based on markers
   `// Each` $r_i$ `preserves imports and namespaces from` $\mathcal{C}$

```
// Stage 2:  Judge (Semantic Filtering)
```
**foreach** *raw statement* $r_i$ **do**
   $y \leftarrow \text{LLM}(r_i)$ ;                                `// Ignore syntax, judge math`
   **if** $y$ *predicts* `correct` **then**
      Add $r_i$ to queue $\mathcal{Q}_{\text{formalize}}$

```
// Stage 3:  Formalizer (Syntax Repair)
```
**foreach** *candidate* $s_{raw} \in \mathcal{Q}_{formalize}$ **do**
   $S \leftarrow s_{raw}$ ;
   $H \leftarrow \emptyset$ ;                                `// Initialize history`
   **for** $t = 1$ **to** $T_{repair}$ **do**
      $E \leftarrow \text{LEANSERVER.CHECK}(S)$ ;                `// Check` $S$ `with imports`
      **if** $E$ *is empty* **then**
         Add $S$ to $\mathcal{Q}_{prover}$ ;                `//` $S$ `is well-formed`
         **break**
      $H \leftarrow \text{Append}(H, S, E)$ ;                `// Record failure & errors`
      $S \leftarrow \text{LLM}(H)$ ;                `// Repair syntax`

```
// Stage 4:  Prover (Proof Generation & Repair)
```
**foreach** *statement* $S \in \mathcal{Q}_{prover}$ **do**
   $H \leftarrow \emptyset$ ;
   **for** $k = 0$ **to** $K_{repair}$ **do**
      $P \leftarrow \text{LLM}(S, H)$ ;                `// Generate full lemma w/ proof`
      $E \leftarrow \text{LEANSERVER.CHECK}(P)$ ;   `// Check if` $P$ `is a valid proof of` $S$ `(with imports)`
      **if** $E$ *is empty* **and** $\text{PROOFBYPASSSCREEN}(P)$ **then**
         **Yield** verified pair $(S, P)$ **break**
      $H \leftarrow \text{Append}(H, P, E)$ ;                `// Feedback:  Code + Error Msgs`

---

## A. Implementation Details and Pseudocode

This appendix provides the precise execution logic of our pipeline (Algorithm 1) and documents the runtime configurations used in our experiments to ensure reproducibility.

### A.1. Pipeline Pseudocode

Algorithm 1 details the interaction logic between the four modules, the persistent queues, and the Lean server.

### A.2. Experimental Configurations

**Methodological Constraints.** We do not fine-tune or update the parameters of any model. All adaptations are performed at inference time via (i) stage-specific prompts, (ii) marker-based structured output extraction, (iii) decoding controls (e.g., temperature/top-$p$), and (iv) Lean-in-the-loop generate–verify–repair using compiler diagnostics. Crucially, we use Lean solely for verification and error feedback; **no external retrieval (e.g., RAG or code search) is used during evaluation**. This setting strictly evaluates the model's internalized mathematical knowledge and reasoning capabilities.

**Overview and Environment.** Our pipeline is implemented in Python, utilizing standard libraries for asynchronous job management and LLM interaction. Key dependencies include `weave` (0.52.17) and `wandb` (0.23.0) for experiment tracking, `persist-queue` (1.1.0) for robust inter-module message passing, and `vllm` for serving open-weight models. The Lean 4 verification backend consists of a customized Lean server wrapper exposing an HTTP endpoint, running on Lean `v4.25.0-rc2`. All candidates are validated by the Lean 4 kernel: statements must type-check, and proofs (when present) must pass kernel proof checking. We interact with the Lean server in a stateless manner to ensure reproducibility. For each verification request, we construct a self-contained Lean file (including necessary imports and open namespaces) and send the full context to the server. This isolates the verification of each candidate, preventing state pollution from previous failed attempts or other lemmas. For statement-only candidates, acceptance requires no error-level diagnostics during elaboration/checking. For proof artifacts, acceptance additionally requires passing the proof-bypass screen defined in Section 3.2.4. To accommodate complex elaboration and compilation tasks, we configure the Lean client with a generous timeout (`timeout=6000` seconds) to prevent premature termination of valid proofs.

The released artifact is version-pinned by the Lean toolchain and Mathlib manifest: all checks are performed under Lean v4.25.0-rc2 and a fixed Mathlib revision. The repository is configured with `relaxedAutoImplicit = false` and Mathlib's standard linter set, and the benchmark/proof files are organized as standalone topic slices rather than as a dense custom import graph, allowing future refreshes to be rechecked locally as Lean/Mathlib evolves.

**LLM Inference Configuration.** We utilize OpenAI-compatible chat completion endpoints for all LLM-backed modules. Unless otherwise specified, we employ the following default settings to ensure sufficient capacity for mathematical reasoning:

- **Token budget:** We set `max_completion_tokens` to 50,000 across all stages to allow for long intermediate reasoning and proof generation.

- **Sampling parameters:** For the Discovery stage, we use exploratory sampling (`temperature=1.0`, `top_p=1.0`). For the Prover stage, we adopt more conservative settings for open-weight models (e.g., `temperature=0.6`, `top_p=0.95`) to reduce hallucinations, while relying on default parameters for frontier models (e.g., GPT-5.1) unless specific reasoning efforts are required (e.g., `reasoning_effort="low"`).

**Stage-Specific Hyperparameters.** We detail the exact logic and runtime configurations for each stage below.

- **Discovery (Generation):** We input the *full text* of the seed Mathlib file into the model context without truncation. We typically run this stage with high concurrency (e.g., 64 threads) using GPT-5.1. The extraction logic relies on the marker `"brainstormed mathlib lemmas"`.

- **Splitting:** Raw generations are split into individual lemma snippets using regex matching on `lemma` or `theorem` keywords. Critical context, including module-level imports and open namespaces, is preserved for each snippet to maintain compilation validity.

- **Judge (Semantic Filtering):** We utilize GPT-5.1 with default sampling parameters and `max_tokens=50,000`. This module runs with high concurrency (up to 100 threads). We use a precise parsing rule: a candidate is accepted if and only if the last non-empty line of the model's response starts with `"correct"` (case-insensitive).

- **Formalizer (Repair Loop):** We employ GPT-5.1 (default sampling, `max_tokens=50,000`) to perform syntax repair. The repair loop runs for a maximum of $T = 10$ trials with a concurrency of 30 workers. In each iteration, the prompt receives the full history of the failed code combined with compiler error messages. We enforce a strict filter: candidates are rejected if the Lean server returns any diagnostics with `severity="error"` or if the error message indicates a duplicate declaration (e.g., containing "has already been declared"). The extraction marker is set to `"error-free code"`.

- **Prover (Proof Search):** For our main benchmarks, we allow up to $K_{\text{repair}} = 2$ repair rounds after the initial proof attempt. The extraction marker is `"### Complete Lean 4 Proof"`. To ensure fair evaluation across different backends (e.g., Goedel, DeepSeek), we inject necessary environment configurations (such as `import Aesop`, `open BigOperators`, and `set_option maxHeartbeats 0`) when required by specific models. We also employ a triviality filter using `aesop`: lemmas that can be solved immediately by replacing `sorry` with the `aesop` tactic (using default configuration and rule sets) are flagged as trivial.

**Artifacts and Data Management.** We use file-based persistent queues to manage state, ensuring that the pipeline can resume from interruptions without data loss. The artifact directory structure follows the pipeline stages: `MathlibLemma` (raw candidates), `MathlibLemmaCorrect` (judged), `MathlibLemmaCompilable` (type-checked statements), and `MathlibLemmaProved` (verified proofs).

## B. Prompt Templates and Extraction Logic

This appendix documents the exact prompt templates used by the **Discovery**, **Judge**, **Formalizer**, and **Prover** modules. To ensure reproducibility, we detail not only the prompt text but also the runtime variable instantiation and the extraction logic.

### B.1. Discovery Module (Generation)

**Input Specification:**

- {`LEAN_FILE`}: A single Mathlib source file corresponding to a specific topic. It typically contains the core definitions and existing lemmas for that topic.

- {`MARKER`}: The extraction marker, set to `brainstormed mathlib lemmas`.

```
{LEAN_FILE}
The lean file above contains many basic lemmas, but not all of them. Many are still
missing. I am frequently surprised that some very obvious lemmas are not formalized yet.
Try to figure out those lemmas. You can guess based on the names of existing ones or try
to make analogues of existing ones. When you brainstorm the lemmas, think not only about
the topic of the file itself but also about how it may interact with topics from other
files. You will write the statements of the lemmas in Lean. Do not try to prove the
lemmas at all; just put a sorry. Generate at least 100 lemmas. Make sure the lemmas you
generate are diverse and not just repeating each other. Make sure they are not merely
renamings of existing lemmas in Mathlib. Make sure they cannot be trivially proved by
just using simp, trivial, or aesop. You can sketch your plan for brainstorming lemmas
first in natural language. After your thinking process, start a new line and write down
"{MARKER}". You will then start with "import Mathlib" and do not add any other imports.
Do not add a namespace or section. Then write down the lemmas you figure out.
```

**Post-processing:** We strip any non-code preamble and extract the Lean snippet appearing after the marker line; if code fences are present, we keep the last fenced Lean block.

### B.2. Judge Module (Semantic Correctness)

**Input Specification:** {`LEAN_STATEMENT`} is instantiated with a single-lemma Lean snippet. The snippet usually contains `sorry` and may not yet be compilable.

```
{LEAN_STATEMENT}
You are given a statement in Lean. Use your best judgement to decide whether this
statement is mathematically correct. You do not need to care whether the Lean syntax is
correct or whether you can prove it in Lean. Outline your rationale. After your thinking
process, at the end of your response, start a new line and use 'correct' or 'wrong' to
indicate your assessment.
```

**Decision Rule:** The candidate is accepted if and only if the *last non-empty line* of the response starts with `correct` (case-insensitive).

### B.3. Formalizer Module (Iterative Repair)

**Input Specification:**

- {`HISTORY`}: A running transcript accumulating pairs of (previous code, compiler diagnostics).

- {`MARKER`}: The extraction marker, set to `error-free code`.

```
{HISTORY}
Try your best to fix the errors. You are using Lean 4. Do not overthink. Try to add
missing stuff first in the errors. Never try to prove the lemma. There is no need to
prove it; just put a sorry. Focus on making the lemma statement compilable. Do not make
any assumption on the environment variables. Make sure the lemma statement is
self-contained. You can assume every related file is already imported, so do not add any
new import. However, the namespace may not be opened properly. At the end of your
response, start a new line with "{MARKER}" and then write down the error-free lean code.
```

**Loop Protocol:** We extract the code following {MARKER}. If compilation fails, the error messages are appended to {HISTORY} for the next turn ($t \leftarrow t + 1$). The loop terminates on success (zero errors) or upon reaching budget $T$.

### B.4. Prover Module (Proof Generation)

**Input Specification:**

- {LEMMA_WITH_SORRY}: The type-checked lemma statement.

- {HISTORY}: Transcript of previous failed proof attempts and errors.

- {MARKER}: The extraction marker, set to ### Complete Lean 4 Proof.

```
{LEMMA_WITH_SORRY}
Complete the proof in Lean. Below is your previous failed trials and the corresponding
errors.
{HISTORY}
You are using Lean 4. Make sure you use correct API and make sure the solution you
generate can compile. You can assume every related file is already imported, so do not
add any new import. However, the namespace may not be opened properly.
Before producing the Lean 4 code to formally prove the given theorem, provide a detailed
proof plan outlining the main proof steps and strategies. The plan should highlight key
ideas, intermediate lemmas, and proof structures that will guide the construction of the
final formal proof. At the end of your response, start a new line with "{MARKER}" and
then write down the error-free lean code with complete proof.
Write the complete lemma with proof including everything you see at the beginning of this
message. Do not just write the proof.
```

**Verification Rule:** A proof is accepted if and only if the self-contained Lean file returns no diagnostics of severity error and the extracted model-generated proof code passes the proof-bypass screen. The screen strips Lean comments and rejects incomplete-proof placeholders (sorry, admit, sorryAx) and model-introduced top-level axiom, constant, or opaque declarations.

### B.5. Robust Output Extraction

To handle the stochastic nature of LLM outputs, we enforce a deterministic extraction pipeline:

1. **Marker splitting:** We locate the *last occurrence* of the stage-specific marker (case-insensitive) to ignore any preliminary "chain-of-thought" or conversational filler.

2. **Code block parsing:** From the text following the marker, we look for fenced code blocks (```lean ... ```). If found, we extract the content of the *last* code block. If no fences are found, we take the raw text.

This ensures that the system reliably captures the final "clean" code intended by the model.

# C. Benchmark Statistics and Detailed Results

## C.1. Evaluation Metrics

We report the **Success@$t$** metric, defined as the cumulative percentage of instances solved using **at most** $t$ repair turns (where $t \in \{0, 1, 2\}$):

- **Success@0 (One-shot):** Success in the initial generation (0 repairs).

- **Success@1:** Cumulative success after up to 1 repair round.

- **Success@2 (Total):** Cumulative success after up to 2 repair rounds. This corresponds to the total system performance reported in our figures.

## C.2. Performance Breakdown by Domain

We provide the fine-grained performance breakdown for each domain in Figures 3, 4, and 5. These figures detail the success rates at each stage of the repair loop (Success@0, Success@1, Success@2), highlighting the impact of iterative refinement.

*Figure 3.* **Performance on Foundational Domain.** This domain comprises standard mathematical structures (e.g., lists, real analysis basics) that are well-represented in the training data. Here, "GPT" denotes GPT-5.1, "GPT-Reasoning" denotes GPT-5.1 with low reasoning, "DS-32B" denotes DeepSeek-R1-Distill-Qwen-32B, and "DS-70B" denotes DeepSeek-R1-Distill-Llama-70B. Goedel-Prover shows the strongest Foundational-domain performance, with Success@2 above 27%. The significant contribution of "0-trial" (One-shot) success indicates that frontier models have internalized many foundational definitions, and general-purpose open-weight models (e.g., DeepSeek) struggle with basic formalization, with scaling from 32B to 70B yielding no performance gain.

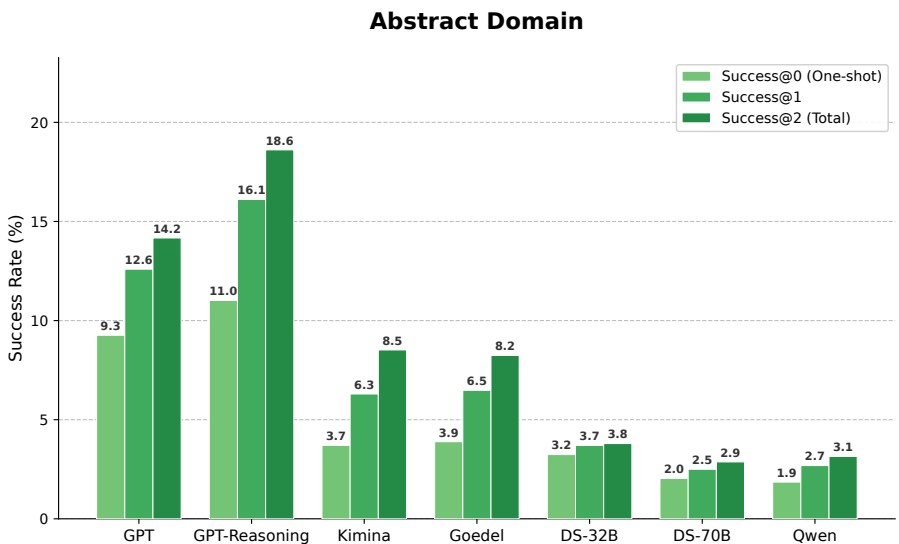

*Figure 4.* **Performance on Applied Domain.** This domain includes fields like probability and information theory, where intuitive concepts often require complex type-class constraints. This domain generally depresses success rates relative to the Foundational domain and remains challenging across models, but models with stronger reasoning, most notably GPT-Reasoning, perform comparatively better.

*Figure 5.* **Performance on Abstract Domain.** This domain covers category theory and differential geometry. Specialized library models such as Goedel degrade sharply relative to the Foundational domain, while GPT-5.1 and GPT-Reasoning remain the strongest models in this domain.

### C.3. Pass@16 Evaluation for a Small Lean Prover

To complement the sequential Success@K evaluation in the main text, we additionally evaluate Goedel-Prover-V2-8B under a standard Pass@16 protocol. These Pass@16 numbers should be interpreted as a small-model independent-sampling stress test, not as a direct comparison with Table 3. Table 3 reports sequential repair-loop Success@2 for Goedel-Prover-V2-32B and other models, whereas this appendix uses the much smaller 8B variant with 16 independent attempts and no repair feedback. Thus, the experiment asks whether repeated parallel sampling from a small Lean-specialized prover already closes a substantial part of the benchmark. For consistency with the main benchmark, we apply the same triviality filter used in Section A.2, excluding statements that are solved immediately by `aesop`. The resulting evaluation set contains 4,028 non-trivial type-checked statements.

*Table 4.* Pass@16 results for Goedel-Prover-V2-8B on the non-trivial MATHLIBLEMMA benchmark.

| Domain | Solved | Total | Pass@16 |
|---|---|---|---|
| Foundational | 1 | 1809 | 0.06% |
| Applied | 7 | 1139 | 0.61% |
| Abstract | 5 | 1080 | 0.46% |
| Overall | 13 | 4028 | 0.32% |

The results remain uniformly low, suggesting that the benchmark is challenging even when a small Lean-specialized prover is allowed multiple independent proof attempts per statement. Together with the main Success@2 results, this supports the same qualitative conclusion: MATHLIBLEMMA is not saturated either by short sequential repair from stronger provers or by independent sampling from a small specialist prover.

## D. Supplementary Details for the Validity Audit

**Failure modes among model-unsolved residuals.** In Section 4.3, we reported that a minority of the audited instances remained unproven even by human experts. To understand the root causes of these failures, we conducted a deep-dive analysis on all 31 audited instances that remained unproven by human experts. Our analysis reveals that failures typically fall into three distinct categories, with **Missing Hypotheses** being the dominant factor (accounting for 81% of failures).

1. **Missing hypotheses or underspecification (25/31, 81%).** These statements capture a plausible mathematical intuition but omit necessary preconditions or dependencies. Common omissions include non-emptiness, non-degeneracy, integrability, measurability, continuity, finite-dimensionality, or explicit relationships between objects that are mathematically linked but introduced as independent Lean parameters.

2. **False as stated (5/31, 16%).** These statements appear to be genuinely false under Lean's precise interpretation, often because of degenerate default cases, incompatible typeclass assumptions, or an overgeneralization of a fact that only holds under additional structure.

3. **Hard technical formalization gap (1/31, 3%).** One instance appears to express a plausibly true mathematical fact, but the required formal proof would require substantial auxiliary development or a more careful reformulation beyond the scope of the audit.

This breakdown highlights that the primary challenge for LLMs is not "imagining false math", but rather **mastering the strict accounting of conditions and dependencies** required by formal systems.

**Held-out residual provability check.** The 138-instance audit in Section 4.3 is our main manual estimate of residual validity. To check that this estimate was not an artifact of that particular seed-stratified sample, we performed an additional held-out audit on five seed files. From each seed file, we selected up to 20 non-trivial type-checked statements that compiled but were not proved by any evaluated prover under the Success@2 protocol. Expert formalizers then attempted to prove these statements in the same open-book setting as Section 4.3. They proved 79 of 94 statements (84.0%). The mean per-seed success rate was 83.7%, with sample standard deviation 4.6%; a Wilson 95% confidence interval for the pooled rate is 75.3–90.1%. This held-out check is not pooled with Table 2, since it uses a different seed-level sampling design. Instead, it serves as a robustness check: it is statistically compatible with the main residual audit and supports the conclusion that many model-unsolved, type-checked statements are provable rather than hallucinated.

## E. Merged Contributions

This appendix presents the three generated lemmas that were merged into Mathlib after human curation. Table 5 gives the upstream pull-request identifiers, merge commits, and Mathlib file locations. Each example below is shown with its mathematical role and the corresponding Lean statement/proof, modulo the surrounding Mathlib imports and namespace context.

*Table 5.* **Merged Mathlib contributions.** The three generated lemmas that were curated by the authors and merged into Mathlib.

| Area | Declaration | PR | Merge commit | Mathlib file |
|---|---|---|---|---|
| Probability | `centralMoment_congr_ae` | #31985 | d7c53e9 | `Mathlib/Probability/Moments/Basic.lean` |
| Measure theory | `Kernel.restrict_const` | #32167 | f157c2c | `Mathlib/Probability/Kernel/Basic.lean` |
| Analysis | `gronwallBound_mono` | #32170 | 1608899 | `Mathlib/Analysis/ODE/Gronwall.lean` |

**Analysis (`gronwallBound_mono`).** This lemma states that the Gronwall-bound expression used in the surrounding Mathlib development is monotone in the time argument, under nonnegativity assumptions on its parameters. The result is mathematically routine, but it packages a step that would otherwise require unfolding the definition of the bound and re-proving monotonicity from the exponential expression. Such monotonicity facts are standard in Gronwall-based estimates, including stochastic approximation arguments; closely related reasoning appears, for example, in Borkar (2009).

*Listing 1.* `gronwallBound_mono`.

```
/-- The Gronwall bound is monotone in time. -/
lemma gronwallBound_mono {δ K ε : ℝ} (hδ : 0 ≤ δ) (hε : 0 ≤ ε) (hK : 0 ≤ K) :
    Monotone (gronwallBound δ K ε) := by
  intro x₁ x₂ hx
  unfold gronwallBound
  split_ifs with hK₀
  · gcongr
  · have hK_pos : 0 < K := by positivity
    gcongr
```

**Measure theory (`restrict_const`).** This lemma states a compatibility property between constant kernels and target-set restriction. A constant kernel ignores its input and always returns the same target measure. After restricting such a kernel to a measurable target set, the result is still a constant kernel, with the original target measure restricted to that set. This is a small API bridge between constant kernels and measure restriction, allowing later proofs to rewrite kernel expressions directly rather than unfold the definitions.

*Listing 2.* `restrict_const`.

```
/-- The restriction of a constant kernel to a measurable set is equal
to the constant kernel of the restricted measure. -/
theorem restrict_const {μ : Measure β} (hs : MeasurableSet s) :
    (Kernel.const α μ).restrict hs = Kernel.const α (μ.restrict s) := by
  ext a
  simp [Kernel.restrict_apply, Kernel.const_apply]
```

**Probability (`centralMoment_congr_ae`).** This lemma states that central moments are invariant under almost-everywhere equality. More precisely, if two real-valued random variables $X$ and $Y$ are equal $\mu$-almost everywhere, then their central moments of the same order with respect to $\mu$ are equal. This is a standard probabilistic congruence principle: quantities defined by integration should depend only on the almost-everywhere equivalence class of the random variable, not on its values on a null set.

*Listing 3.* `centralMoment_congr_ae`.

```
/-- Central moments are equal for almost-everywhere equal random variables. -/
lemma centralMoment_congr_ae {X Y : Ω → ℝ} (hXY : X =ᵐ[μ] Y) :
  centralMoment X p μ = centralMoment Y p μ := by
simp only [centralMoment, integral_congr_ae hXY]
refine integral_congr_ae ?_
filter_upwards [hXY] with x hx using by simp [hx]
```

# F. Prompt Ablation: Explicit Necessary Conditions

Motivated by the missing-hypotheses failure mode, we ran a small prompt ablation in which the Discovery module was instructed to explicitly list the necessary hypotheses before producing each formal statement. In this pilot, 123/160 generated

statements compiled. Manual review of all 123 compilable statements found that 117 were provable as stated; among the six residual failures, only two appeared to be genuine missing-hypothesis cases. For comparison, an old-prompt run on the same scale also produced 123/160 compilable statements; manual review found seven residual unresolved cases, of which two appeared to be genuine missing-hypothesis cases. Thus, explicitly prompting for necessary conditions is reasonable and may slightly improve precision on this pilot, but among compilable statements it did not substantially reduce missing-hypothesis failures in this small pilot. We therefore leave a systematic prompt redesign to future work.

