# OpenReview forum: "MathlibLemma: Folklore Lemma Generation and Benchmark for Formal Mathematics"
_ICML.cc/2026/Conference — ICML 2026 regular_

### Official Review · Reviewer_79R3 · 2026-03-04

**Soundness:** 2
**Presentation:** 2
**Significance:** 2
**Originality:** 2
**Overall Recommendation:** 2
**Confidence:** 4

**Summary:**

This paper presents MathlibLemma, a four-agent LLM pipeline (Discovery, Judge, Formalizer, Prover) that generates candidate "folklore lemmas" from Mathlib seed files, filters and repairs them via compiler feedback, and attempts proofs. The pipeline produces a benchmark of 4,028 type-checked Lean 4 statements. Seven SOTA models are evaluated; the best individual model solves 22.32% and the union reaches 44.99%. A human audit finds 78% of unsolved residuals are provable. Three lemmas have been merged into Mathlib.

**Compliance With Llm Reviewing Policy:**

Affirmed.

**Final Justification:**

1. The Mathlib curation concern (Q2) is adequately addressed.
2. The downstream ablation remains missing; the authors' scope narrowing from LLM proving to human formalization contradicts the introduction's own motivation rather than resolving the evidence gap.
3. The distinction from generic conjecture generation remains empirically unvalidated, resting on prompt design rather than measurable output differences.

Since (2) and (3) concern the paper's core claims, my score remains unchanged.

**Key Questions For Authors:**

1. The paper's core motivation is that missing folklore lemmas impede proof development. Can you provide any evidence that the generated lemmas actually improve downstream proving? For instance, do LLM provers achieve higher success rates on existing Mathlib theorems when your library is available as auxiliary context? This is the most natural experiment suggested by your own motivation (Section 1), and a positive result would substantially strengthen the paper.

2. One concern I'd like to raise for discussion: Mathlib appears to be intentionally curated for structural significance, deliberately excluding many "obvious" consequences to maintain coherence and navigability. If so, the absence of folklore lemmas may be a design choice rather than a deficiency. Have the authors considered whether the generated folklore should exist as a *separate, searchable auxiliary layer* rather than as Mathlib contributions? More broadly, what integration architecture do you envision for making these lemmas practically accessible to provers and human formalizers at scale? I think engaging with this question could significantly strengthen the paper's practical narrative.

**Limitations:**

The authors acknowledge deduplication, semantic drift, and the gap between verified proofs and mergeable contributions. However, two more fundamental limitations are not discussed: (1) the absence of any downstream evaluation of whether generated lemmas are actually useful for proof development, and (2) the question of how a large-scale folklore library should be architecturally integrated into the Lean ecosystem without compromising Mathlib's intentional curation standards.

**Strengths And Weaknesses:**

### Strengths

1. The last-mile barrier problem is genuine, well-motivated, and practically important. The examples from Tao's PFR project and others are convincing.

2. The pipeline design is sensible, including Discovery Agent, Judge Agent, Formalizer Agent and Prover Agent is well-justified. The funnel statistics (Table 1) and failure taxonomy (Appendix D) provide useful empirical insights.

3. The prover evaluation is thorough and yields interesting findings: the specialist-vs-generalist trade-off (Goedel strong on foundational, weak on abstract), and the striking orthogonality across models (union 2× best individual).

### Weaknesses

1. **No evidence that generated lemmas are useful.** The paper's stated goal is closing the last-mile barrier that limits Lean's usability. But the evaluation only shows the pipeline produces *true and non-trivial* statements—never that these statements are *useful*. This disconnect pervades the entire contribution. The Discovery Agent asks an LLM to guess what might be missing, but nothing validates that outputs correspond to actual gaps encountered during proof development. "Novel" and "non-trivial" (not solved by `aesop`) does not imply "needed." The benchmark then evaluates whether LLM provers can *prove* these statements: a standard ATP benchmark that measures prover capability, not generation quality. Provability by an LLM is orthogonal to whether a lemma fills a real library gap. The natural experiment is missing: give provers access to the generated lemma library as auxiliary context, then measure whether they solve more theorems on existing proving tasks. The paper itself motivates this exact scenario (Section 1: "When a necessary lemma is missing, an LLM cannot simply invoke it via a concise reference. Instead, it is forced to reconstruct the result from scratch, transforming a single inference step into a lengthy derivation."), yet never tests it.

2. **Limited technical novelty.** Each pipeline component (LLM generation with structured prompts, LLM-as-a-judge, compiler-in-the-loop repair, multi-trial proof search) is individually standard in recent formal theorem proving literature. The claimed novelty rests on the "folklore mining" framing, but since the distinction from generic conjecture generation (LeanConjecturer, Lemmanaid) is never empirically validated, this remains a rhetorical rather than technical contribution.

---

> ### Author Rebuttal · Authors · 2026-03-30
>
> Thank you for the thoughtful review. We believe the key clarification is the scope of our contribution. Our claim is not that each pipeline component is individually novel; rather, we formulate a new task, namely AI-assisted mining of library-worthy folklore lemmas from an existing formal library, and show that this task is feasible in practice. In addition, other reviewers viewed this task as novel, genuine, and practically interesting, and several highlighted the benchmark findings, especially the specialist-vs-generalist tradeoff and the strong complementarity across provers, as interesting in their own right. Our paper therefore has two contributions: a pipeline for folklore-lemma mining, and a benchmark that exposes nontrivial differences among current formal reasoning systems.
>
> > W1 / Q1 / L1. Usefulness and missing downstream ablation
>
> We agree that the strongest direct test would be a downstream ablation: giving a prover access to the generated lemma library and measuring whether this improves success on existing proving tasks. We did not run that experiment in this paper, and we will acknowledge this limitation more explicitly in the revision. In practice, such an evaluation requires a full retrieval/agent stack, and for frontier closed models the most realistic setup would use expensive agent-style APIs (e.g., Codex/Claude Code-style systems), which was beyond our budget for this submission.
>
> That said, we do have nontrivial evidence of usefulness. We turned 6 generated lemmas into Mathlib PRs; 3 have already been merged, 1 was judged redundant, and we did not continue the remaining 2 due to time constraints. We intentionally used this as a small pilot rather than attempting bulk submission of all roughly 2,000 proved lemmas. Mathlib is maintained by volunteers, and directly submitting AI-generated PRs at that scale would impose substantial review burden without prior coordination. For any larger-scale integration effort, our plan is to first discuss with maintainers what curation and submission process would be most useful and responsible. Thus, we use selective upstreaming as a high-standard validation signal rather than a bulk-ingestion mechanism.
>
> We also have a concrete qualitative example. One merged lemma establishes monotonicity of a Grönwall-type time bound, a reusable intermediate step that appears in classical stochastic approximation arguments; for example, closely related reasoning appears at the beginning of p.14 of [2], a standard textbook in stochastic approximation. We agree, however, that a controlled quantitative downstream study would further strengthen the paper, and we will state this more clearly in the revision.
>
> Ref. [2] Borkar, Vivek S. Stochastic approximation: a dynamical systems viewpoint. Vol. 100. Cambridge: Cambridge University Press, 2008.
>
> > W2. Limited technical novelty
>
> We agree that the ingredients of our pipeline, including LLM generation, filtering/judging, repair, and proof search, are not novel in isolation. Our novelty lies at the task level and in the end-to-end target. We study a stricter problem than generic conjecture generation: not just producing true statements, but discovering machine-checked lemmas that are realistic candidates for integration into an existing library. The contribution is therefore a minimal viable pipeline for a new library-improvement task, together with real-world validation via Mathlib PRs, including three merged lemmas. To our knowledge, prior AI-generated theorem/proof work has not demonstrated this kind of upstream integration into Mathlib.
>
> > Q2 / L2. Curation and integration architecture
>
> Thank you for raising this broader integration question. We agree that, at scale, it is important to think carefully about how folklore lemmas should be organized and exposed to both provers and human formalizers. However, to our knowledge, Mathlib’s official contribution guidelines do not state that simple theorems are undesirable per se. In practice, many accepted PRs are relatively small: in our interaction-log data, 37% of merged PRs have total diff size at most 10 lines, and 52% have total diff size at most 50 lines. We therefore believe it is too strong to interpret the absence of folklore lemmas as generally reflecting a deliberate policy of exclusion.
>
> In our current paper, Section 6 (line 402-408) already positions the verified library as an immediately usable reference database for human formalizers, in addition to being a staging area for selective upstreaming into Mathlib. That is also our intended practical picture: a broader standalone folklore layer, with selective upstreaming for the subset that meets Mathlib’s standards and is judged sufficiently reusable. We will revise the paper to make this scope and integration story clearer.
>
> We will revise the paper to make these scope claims and limitations clearer.

---

> > ### Author Rebuttal · Reviewer_79R3 · 2026-04-04
> >
> > Thank you for the rebuttal. The Q2 response on Mathlib curation is reasonable.
> > However, my core concerns remain unresolved. On W1, the rebuttal concedes that the downstream ablation is missing, and the offered substitutes do not close the gap: the benchmark measures whether provers can prove the generated lemmas, not whether the lemmas improve downstream proving. On W2, the claim that the Discovery Agent targets structural gaps rather than generic conjectures remains empirically unvalidated. I will keep my score unchanged.

---

> > > ### Author Response · Authors · 2026-04-06
> > >
> > > > W1.
> > >
> > > We thank the reviewer for the continued engagement. We agree our original wording did not make the scope sufficiently clear. Our primary target is assisting human formalization rather than improving LLM prover performance. The brief discussion in the introduction about missing lemmas regrading LLM-based reasoning was intended only as motivation, not as an empirical claim that our generated lemmas improve downstream proving. We acknowledge that this wording was misleading, and in the revision we will remove revise it to make the scope explicit.
> > >
> > > Under this clarified scope, we agree that a downstream ablation, i.e., giving provers access to the generated library and measuring effects on existing tasks, would be a valuable extension. However, under the clarified scope of the paper, we see it as a valuable extension rather than necessary evidence for the narrower claims we make here.
> > >
> > > We highlight three signals for the practical value of our work.
> > > - **Mathlib acceptance.** Three generated lemmas have been merged into Mathlib by human maintainers, which is a strong external signal of mathematical usefulness and library quality.
> > > - **Benchmark sustainability and ease of updating.** MiniF2F and ProofNet [1, 2] remain pinned to Lean 3 commits now several years behind current Mathlib, and the miniF2F-v2 update [3] required manual review of more than half of MiniF2F’s problems even without a version upgrade. By contrast, our pipeline regenerates the benchmark automatically for new Mathlib snapshots with no manual intervention.
> > > - **Benchmarking a new regime.** Our benchmark targets folklore lemmas in current Mathlib, a regime not previously evaluated; other reviewers found the benchmarking results insightful in their own right, which we take as evidence that this is a meaningful contribution independent of downstream ablations.
> > >
> > > We will add an explicit statement in the paper acknowledging that a controlled downstream ablation would be valuable future work.
> > >
> > > > W2.
> > >
> > > We agree folklore lemma generation is a specific instance of conjecture generation; our claim is not that it lies outside that category, but that it emphasizes a particular regime. Folklore lemmas are not new mathematical truths waiting to be discovered, they are useful intermediate results absent as standalone declarations. Experts consider them obvious, yet Mathlib has never explicitly stated them. For example, "matrix multiplication is linear" is universally known, yet Mathlib requires each concrete instantiation separately: $X(v + u) = Xv + Xu, (v + u)^T X = v^T X + u^T X$, and many variants. Similarly, $\mathbb{E}\[ f(X)(i) | Y\] = \mathbb{E}\[ f(X) | Y\](i)$ (that the i-th coordinate of a conditional expectation of a vector-valued random variable equals the conditional expectation of its i-th coordinate) reads as trivial to a probabilist, yet Mathlib was missing it for a long time; formalizing it requires explicitly identifying conditional expectation as a linear map under the measure-theoretic definition. Such lemmas are invisible to systems looking for new mathematical truths, precisely because they are not new: they are merely unwritten.
> > >
> > > This distinguishes our setting from prior Lean/Mathlib conjecturing work. For example, LeanConjecturer [4] uses rule-based, file-local context extraction and prompts generation from existing declarations, which is well suited to extending or extrapolating from existing declarations, whereas our setting is driven by identifying missing but useful intermediate lemmas. The Conjecturing-Proving Loop [5] iteratively builds on previously proven theorems to rediscover theorems from published mathematical papers, i.e., statements that are already articulated in the literature. Folklore lemmas, by contrast, exist only very implicitly in the literature. These are closely related directions, but they emphasize different parts of the conjecture-generation space.
> > >
> > > Our design choices reflect this regime: an absence-oriented discovery prompt asks the model to identify obvious but unformalized lemmas; a "make analogue of existing ones" instruction targets concrete instantiations of the same abstract result; and an explicit instruction to consider imported files and neighboring theories guides reasoning about what is missing.
> > >
> > > Ref.
> > > [1] Zheng, Kunhao, Jesse Michael Han, and Stanislas Polu. "Minif2f: a cross-system benchmark for formal olympiad-level mathematics." ArXiv Preprint (2021).
> > >
> > > [2] Azerbayev, Zhangir, et al. "Proofnet: Autoformalizing and formally proving undergraduate-level mathematics." ArXiv Preprint (2023).
> > >
> > > [3] Ospanov, A., Farnia, F., & Yousefzadeh, R. miniF2F-Lean Revisited: Reviewing Limitations and Charting a Path Forward. ArXiv Preprint (2025).
> > >
> > > [4] Onda, Naoto, et al. "LeanConjecturer: Automatic Generation of Mathematical Conjectures for Theorem Proving." ArXiv Preprint (2025).
> > >
> > > [5] Kasaura, Kazumi, et al. "Discovering New Theorems via LLMs with In-Context Proof Learning in Lean." ArXiv Preprint (2025).

---

### Official Review · Reviewer_jm7V · 2026-03-13

**Soundness:** 2
**Presentation:** 2
**Significance:** 3
**Originality:** 3
**Overall Recommendation:** 4
**Confidence:** 3

**Summary:**

This paper tackles the "folklore lemma" problem by filling missing gaps in formal math libraries. Focusing on Lean 4’s Mathlib, the authors introduce a paradigm for generating adjacent lemmas to power future formalization/discovery. Their benchmark successfully identifies new lemmas that have already been merged into Mathlib.

**Compliance With Llm Reviewing Policy:**

Affirmed.

**Final Justification:**

After a series of discussions with the authors, I will maintain my initial positive assessment. I believe that this work is of interest to the community and will further strengthen open-source formal math libraries.

**Key Questions For Authors:**

- What safeguards are in place beyond the judgment of novelty and the correctness of the generated lemmas?
- How do authors ensure the longevity of the proposed benchmark, especially since this pipeline must adapt to future Lean releases?
- Could you provide Pass@16 results for smaller dedicated provers?
- Why are individual MathlibLemma components called "agents", not "modules"?

**Limitations:**

yes

**Strengths And Weaknesses:**

## Strengths
- **Strong motivation**. The paper addresses the significant task of automated lemma discovery. While previous research has shown that the current generation of provers can solve complex Olympiad-level problems, this work provides a natural extension to more practical settings, demonstrating how LLMs can help formalize and integrate mainstream mathematics into formal libraries.

- **Intuitive design**. The work presents a clean, intuitive lemma generation pipeline that incorporates multiple safeguards. The core approach is adaptive and could be adopted by larger communities beyond formal math.

- **Contribution to Mathlib**. The authors demonstrate that this method has already contributed to Mathlib and operates in a semi-automated fashion. Such work encourages further efforts to use current technology in bridging the gap between formal mathematics and the broader math community.

With that said I would like authors to address certain points.

## Weaknesses
- **Concerns regarding the Formalizer agent**. Upon reading the manuscript, I noticed that the current pipeline only checks for compilation success, and refinements are primarily targeted toward fixing syntax errors. However, in autoformalization literature, such as [1], researchers often pair the translation layer with back-translation to ensure semantic consistency, i.e., that the formal goal matches the informal problem task. Moving forward, the lack of an additional translation check after each Lean goal modification could contribute to downstream hallucinations. How do the authors ensure that this does not occur? What safeguards are in place beyond the judgment of novelty and the correctness of the generated lemmas?

- **Susceptibility to bugs**. Recent literature suggests that there are certain bugs and Lean features that could potentially compromise the correctness of the benchmark. One notorious Lean feature is inferred variable types, where variables are assigned a type based on the context. This could result in lemmas with limited scope (e.g., a proof only found for $\mathbb{N}$ vs. $\mathbb{R}$). Many such issues can be addressed with explicit variable or function types. How do the authors ensure the longevity of the proposed benchmark, especially since this pipeline must adapt to future Lean releases?

- **Prover agent evaluation**. I understand the intent behind using Success@K instead of Pass@K; however, the industry standard is Pass@16–32 to test the difficulty of a benchmark. To further strengthen the paper, I would suggest including Pass@16 results, at least for the smaller Lean-dedicated provers, such as Goedel-Prover-V2-8B or Kimina-Prover-8B.

- **Minor points to address**. In this section, I have bundled smaller comments regarding this work. (1) There is a typo in Section 3.2.3, line 193: it should read "Kimina Lean server," not "Kinima Lean server." (2) Why are the submodules called "agents" in this work? I understand the appeal, but most of these modules are simply prompted LLMs; the only submodule that could truly be characterized as an agent is the Formalizer Agent. To further improve clarity, I would suggest renaming them as (sub-)modules and treating MathlibLemma itself as the agent.

---

[1] Gao et al. (2025) Herald: A Natural Language Annotated Lean 4 Dataset. ICLR 2025

---

> ### Author Rebuttal · Authors · 2026-03-30
>
> Thank you for the positive assessment. We would like to emphasize that our main contribution is to formulate a new task, namely AI-assisted mining of library-worthy folklore lemmas from an existing formal library, and to show that this task is feasible in practice. Even with a relatively simple end-to-end pipeline, the system can already generate, prove, and in some cases upstream such lemmas. We will revise the paper to make this framing clearer.
>
> > W1 / Q1. Formalizer semantic fidelity and existing safeguards
>
> Thank you for raising this important point. We agree that compilation success does not guarantee semantic fidelity, and this is a real limitation. More generally, reliably verifying semantic preservation after iterative formal edits remains an open problem in autoformalization; we do not claim to solve it here. We will cite [1] as relevant prior work and acknowledge this limitation more explicitly.
>
> Our current safeguards are conservative but limited: candidates are first filtered semantically by the Judge, then required to type-check in Lean, and a small subset that is upstreamed receives additional human review from both the authors and Mathlib maintainers. Because our setting starts from existing formal-library contexts rather than free-form natural language, the semantic gap is narrower than in standard autoformalization, but it is not eliminated.
>
> We also take a conservative deployment approach: the generated collection dataset is primarily treated as a standalone folklore-lemma resource, and only selected lemmas are considered for upstreaming. Of the 6 generated lemmas we turned into Mathlib PRs, 3 were merged, 1 was rejected, and 2 were not pursued further due to timeline constraints. This suggests both that the pipeline can produce useful lemmas and that curation remains necessary.
>
> More broadly, stronger semantic-faithfulness checks, such as back-translation or entailment-style verification after repair, remain open problems in autoformalization, and we do not know how to solve them perfectly at present. Our claim is therefore not that these issues are resolved, but that even under these unresolved limitations, a simple pipeline already demonstrates the feasibility of the proposed new task. We will acknowledge these gaps more explicitly in the revision.
>
> Ref. [1] Gao et al. (2025), Herald: A Natural Language Annotated Lean 4 Dataset, ICLR 2025.
>
>
> > W2 / Q2. Susceptibility to bugs and the longevity of the proposed benchmark
>
> Our guarantee is version-specific: each instance is checked in a fixed Lean/Mathlib environment, and validity is determined by the Lean kernel in that environment. We therefore view the benchmark as a versioned artifact, not a permanently frozen one. Its longevity comes from reproducibility and refreshability: it is tied to a fixed Mathlib snapshot and can be regenerated and rechecked as Lean/Mathlib evolve.
>
> This issue is not unique to our setting; any benchmark built on top of an evolving formal library faces the same maintenance challenge. Some items may also cease to be “missing folklore lemmas” because they are later added to Mathlib; we view this as a natural and positive evolution. We will clarify this in the revision, along with limitations such as inferred types or unintended overspecialization.
>
> > W3 / Q3. Could you provide Pass@16 results for smaller dedicated provers?
>
> Yes. Following your suggestion, we ran Pass@16 for Goedel-Prover-V2-8B on the three benchmark strata reported in the paper. The results are uniformly low: 0.053% on foundational (1/1887), 0.574% on applicational (7/1219), and 0.578% on abstract (7/1211), for a merged Pass@16 of 0.348% (15/4317).
>
> These runs also help diagnose failure modes. Across all three strata, failure is dominated by semantic proof failure rather than formatting or extraction issues: 96.7% of failed files contain at least one Lean-like proof attempt that still fails semantically, while purely format-related failures account for only 0.6% of failed files. There were also no runtime errors in the final applicational/abstract runs. We will add the Pass@16 result and this failure-mode clarification in the revision.
>
> > W4 / Q4. Minor points to address; Why are individual MathlibLemma components called "agents", not "modules"?
>
> Thank you for catching the typo in Section 3.2.3: it should be “Kimina Lean server,” not “Kinima Lean server.” We also agree that “module” is more accurate than “agent” for several internal components. We will revise the terminology accordingly, reserving “agent” for the full MathlibLemma system or for components that genuinely involve iterative tool use.

---

> > ### Author Rebuttal · Reviewer_jm7V · 2026-04-03
> >
> > If the guarantee is version-specific and the authors do not employ any safeguards (even pre-existing ones), I am inclined to decrease my score. There are numerous ways to address version changes and potential issues, such as utilizing explicit variable types or avoiding specific constructions like the introduction of new axioms. If the authors rely solely on LLMs for the heavy lifting and the dataset components become obsolete over time, then what is the value of the contribution? The Lean community releases new versions rapidly, and the pace at which Mathlib and its tactics evolve is accelerating due to increased research interest. These considerations are vital when proposing a new lemma generation and benchmarking framework.
> >
> > How do the authors plan to address these maintenance issues? At this stage, I am inclined to decrease my score.

---

> > > ### Author Response · Authors · 2026-04-03
> > >
> > > Thank you for pressing on this point. We recognize that the core concern is not about permanent compatibility guarantees, but about whether the framework incorporates concrete safeguards and whether its contribution retains value as Lean and Mathlib evolve.
> > >
> > > > ... the authors rely solely on LLMs for the heavy lifting ...
> > >
> > > **We have both Lean-level and structural safeguards in place.**
> > >
> > > 1. At the toolchain level, we configure the build system with `relaxedAutoImplicit = false`, which disables the majority of automatic implicit variable binding according to Lean documentation and ensures that the elaboration behavior of all generated lemmas is explicit and stable.
> > > 2. At the structural level, each lemma in the benchmark depends only on existing results in Mathlib and does not depend on other newly added lemmas. When a Lean or Mathlib upgrade breaks one lemma, the failure is strictly local to that file and does not cascade through a repo-internal import chain, making it straightforward to identify and rerun only the affected subset.
> > > 3. At the structural level, we also have a wide coverage of topics. So even if a Lean and Mathlib upgrade breaks one commonly used API in some topic, many other parts of the benchmark can still survive.
> > >
> > > We apologize for not making these points explicit earlier; we had mistakenly assumed the reviewer was looking for something to guarantee 100% forward compatibility, and so addressed a different question. We will add more discussion about maintenance in the revision.
> > >
> > > That being said, whatever safeguards we use, eventually our dataset (and any Lean repo) will become obsolete over time if no further human intervention is provided. So the reviewer's question below is especially important
> > >
> > > > ... the dataset components become obsolete over time, then what is the value of the contribution?
> > >
> > > **First, our dataset reveals new knowledge about LLM's capability in an area that no existing benchmark reveals.** MiniF2F and ProofNet [1, 2] are collections of pre-existing, named problems sourced from competition archives and undergraduate textbooks respectively. LeanDojo Benchmark 4 [3] takes a different approach, extracting theorems and proofs that already exist and are already proved in Mathlib via automated repository tracing. Our dataset targets a category that none of these cover: folklore lemmas.
> > > Our benchmark thus already reveals new insights about LLM's capability.
> > >
> > > **Second, when a dataset does eventually become obsolete, the cost of regeneration varies greatly.** MiniF2F is pinned to Lean 3.42.1 (released March 2022) and a mathlib3 commit from April 2022, now roughly four years behind current Mathlib. ProofNet is pinned to Lean 3.50.3 (December 2022) and a mathlib3 commit from February 2023, roughly three years behind. Both remain on Lean 3, which is no longer actively maintained by the community. The recent miniF2F-v2 [4] illustrates the cost of upgrading such a dataset concretely: it required manually reviewing and correcting formal statements for more than half of MiniF2F's problems, without even attempting a version upgrade. Our pipeline, by contrast, can regenerate a benchmark against a new Mathlib snapshot by re-running the affected subset automatically, with no manual intervention required.
> > >
> > > **Third, regeneration is valuable even before obsolescence occurs.** As models trained on fixed benchmarks improve, overfitting becomes a legitimate concern. Because existing datasets are static, there is no principled remedy. Our pipeline makes it straightforward to regenerate a fresh snapshot at any time, providing a natural defense against overfitting.
> > >
> > > **Fourth, one intended usage of our pipeline is to upstream discovered folklore lemmas directly into Mathlib.** The three PRs already merged confirm that this lifecycle is viable. Those merged results become permanently community-maintained contributions, retaining their value regardless of the state of our repository or any future version drift.
> > >
> > > We hope this fully addresses the reviewer's concerns. We will incorporate all of the above into the revision.
> > >
> > > Ref.
> > >
> > > [1] Zheng, Kunhao, Jesse Michael Han, and Stanislas Polu. "Minif2f: a cross-system benchmark for formal olympiad-level mathematics." ArXiv Preprint (2021).
> > >
> > > [2] Azerbayev, Zhangir, et al. "Proofnet: Autoformalizing and formally proving undergraduate-level mathematics." ArXiv Preprint (2023).
> > >
> > > [3] Yang, K., Swope, A., Gu, A., Chalamala, R., Song, P., Yu, S., ... & Anandkumar, A. Leandojo: Theorem proving with retrieval-augmented language models. Advances in Neural Information Processing Systems (2023).
> > >
> > > [4] Ospanov, A., Farnia, F., & Yousefzadeh, R.  miniF2F-Lean Revisited: Reviewing Limitations and Charting a Path Forward. ArXiv Preprint (2025).

---

### Official Review · Reviewer_ns2T · 2026-03-13

**Soundness:** 4
**Presentation:** 4
**Significance:** 2
**Originality:** 3
**Overall Recommendation:** 5
**Confidence:** 4

**Summary:**

The paper introduces a multi-agent LLM pipeline (MathlibLemma) to automatically discover and formalize math theorem/lemma/facts that mathematicians "take for granted" but are missing from Lean 4's Mathlib library. The paper collect Mathlib data, through the 4 stages pipline: Discovery agent, Judgement agent, Formalizer agent, Prover agent. The system produces a benchmark of 4,028 type-checked Lean statements and a verified library of 1,812 proved lemmas.

**Compliance With Llm Reviewing Policy:**

Affirmed.

**Final Justification:**

The rebuttal resolved my concerns, raise score to 5 (accept)

**Key Questions For Authors:**

Did you just ask AI oneshot to let it generate "folklore" lemmas?;
3/1812 gap: How many of the 1812 wee send to Mathlib community for reviewing?

**Limitations:**

1. Discovery agent is static one-shot:
The Discovery Agent relies on one-shot prompting from a single seed file, which limits it to gaps visible from that local context. A possible approach can be dynamic by equipping the agent with RAG over the full library, dependency graph traversal, or proof-attempt-driven gap detection (where the agent discovers missing lemmas by encountering them as obstacles during proof search). This would align the discovery mechanism with how experienced contributors actually identify library gaps. The authors mention retrieval-augmented generation as future work (Section 7), but given that it addresses the pipeline's most fundamental limitation, it deserves more discussion here.

2. Cost-effectiveness is unaddressed.
The pipeline runs GPT-5.1 across four stages with up to 10 repair times (Formalizer agent) and 50,000 max tokens per call (Appendix A.2), yet no cost breakdown is reported. This matters because of an internal tension in the paper's framing: folklore lemmas are defined as "obvious" routine facts (in mathematics), yet the best single model only proves 22% of them and even seven frontier models collectively reach only 45% (Table 3). If these lemmas are truly routine, why do they require frontier-level compute? If they require frontier-level compute, are they really folklore? Either way, with only 3 of 1,812 proved lemmas merged into Mathlib, the cost-per-impact ratio appears very high. A dollars-per-merged-lemma estimate would help assess whether this is a scalable methodology or an expensive proof-of-concept.

3. Actual usefulness:
No discussion of whether this pipeline actually saves total human effort compared to just asking experienced Mathlib contributors directly, given that the 1,800 generated proofs still need expert review to upstream.

**Strengths And Weaknesses:**

Strengths:
1. The focus on the "take for granted" math theorems are very interesting; the multi-agent modular design is complete, clean and well-motivated, factorizing semantic filtering, syntactic repair, and proof search into separate stages makes the failure modes transparent.
2. Comprehensive evaluation with interesting findings. The specialist-vs-generalist trade-off is striking: Goedel achieves 29.63% on foundational tasks but drops to 12.96% on abstract math, while GPT-Reasoning is more consistent (Table 3). The union-vs-individual gap (45% vs 22%) suggests strong model orthogonality. The finding that distilled DeepSeek-32B outperforms much larger Qwen-235B (7.05% vs 2.81%) is also noteworthy.

Weakness:
1. Effort-vs-output proportionality: we have ~2k lemmas but only 3 are merged (still congrats on it!). Identifying what lemmas are missing is arguably the cheap part of the problem. A small group of experienced Mathlib contributors could produce a comparable (and likely higher-quality) list in a focused session, where the AI generated list still take expert time to review.
2. From the Appendix B.1, we can see the discovery agent feeds a Mathlib file to GPT-5.1 and asks it to generate "at least 100 lemmas." There is no retrieval-augmented generation, no analysis of Mathlib's dependency graph to identify structural gaps, no comparison against existing lemma databases beyond the soft instruction to "not merely rename existing lemmas."

---

> ### Author Rebuttal · Authors · 2026-03-30
>
> Thank you for the positive assessment and for recognizing the importance of the problem. Our main contribution is to formulate a new task, namely AI-assisted mining of library-worthy folklore lemmas from an existing formal library, and to show that it is already feasible in practice with a simple end-to-end pipeline. We will make this framing clearer in the revision.
>
> > W1 / Q2. 3/1812 gap: No discussion of whether this pipeline actually saves total human effort; How many of the 1812 were send to Mathlib community for reviewing?
>
> Yes. Our claim is more limited: this paper presents a minimum viable framework for large-scale folklore-lemma discovery and proving, shifting human effort from open-ended discovery/proving to targeted review and selection. More importantly, upstreaming is not the only intended use case. As discussed in Section 6 (Line 402-408), the generated proofs can be used directly as a standalone repository of machine-checked folklore lemmas and as an immediately usable reference database for human formalizers. In this use case, users can import the repository into downstream Lean projects and search the AI-generated lemma library directly; this does not require expert review for upstreaming into Mathlib.
>
> We submitted only 6 generated lemmas as Mathlib PRs: 3 were merged, 1 was rejected, and 2 were not pursued further due to timeline constraints. We intentionally did not attempt to submit all roughly 2,000 proved lemmas as PRs. Mathlib is maintained by volunteers, and a bulk submission at that scale would create substantial review burden and would not be a responsible way to engage with the maintainer workflow without prior coordination. For any larger-scale integration effort, our plan is to first discuss with Mathlib maintainers what curation and submission process would be most useful.
>
> > W2 / Q1 / L1. There is no retrieval-augmented generation, no dependency-graph analysis, and no strong comparison against existing lemma databases; discovery agent is static and largely one-shot; did you essentially just ask AI in one shot to generate “folklore” lemmas?
>
> Yes. Our goal is to establish a minimal feasible pipeline and show that even a lightweight discovery setup can already surface genuine, machine-checkable, previously absent folklore-style lemmas at scale. We do not present the current Discovery Agent as a final design.
>
> We agree that stronger repository-aware methods would likely improve both precision and coverage, including retrieval over Mathlib, dependency-graph-guided exploration, proof-attempt-driven discovery, and stronger novelty checking against existing lemma databases. These could be implemented using repository-aware coding-agent APIs such as Codex- or Claude Code–style agents, but doing so at our scale would be substantially more expensive, so we leave this to future work. We will clarify this limitation in the revision.
>
> > L2 / L3. Cost-effectiveness and actual usefulness.
>
> The total API cost was about \\$700: roughly \\$200 for discovery/filtering/formalization, which produced the 4,028 type-checked statements in Table 1, and \\$500 for downstream proving experiments. This is about \\$0.05 per type-checked statement, \\$0.25 per proof success for proving alone, or about \\$0.35 if the full pipeline cost is amortized over the roughly 2,000 proof successes across models.
>
> As stated, we intentionally submitted only a small sample, since mass submission would not be feasible or appropriate for community review. Thus, we use selective upstreaming as a high-standard validation signal rather than as a bulk-ingestion mechanism. Even under a conservative assumption that only 10% of the roughly 2,000 proof successes are ultimately PR-worthy, the implied cost would still be about \\$3.5 per candidate PR-worthy lemma.
>
> We also do not claim that usefulness should be measured only by upstreaming. An important use case is the generated collection as a standalone Lean repository: it is kernel-checked, compiles cleanly, and is filtered to avoid semantic duplication against Mathlib, so it can be imported into downstream projects with little additional overhead even without being merged into Mathlib.

---

> > ### Author Rebuttal · Reviewer_ns2T · 2026-04-01
> >
> > The rebuttal resolved my concerns

---

### Official Review · Reviewer_rjii · 2026-03-17

**Soundness:** 3
**Presentation:** 3
**Significance:** 3
**Originality:** 3
**Overall Recommendation:** 4
**Confidence:** 2

**Summary:**

When formalizing proofs in Lean 4, mathematicians frequently get stuck on small "obvious" facts that are not in Mathlib. These are "folklore lemmas", things every mathematician knows but nobody has bothered to formalize. This paper builds a multi-agent LLM system to automatically discover and prove these missing lemmas. Four agents handle discovery, semantic filtering, syntax repair, and proof generation. The system produces 4,028 type-checked Lean statements, of which 1,812 are proved, and three have been merged into the official Mathlib library. A human audit of 138 sampled unsolved instances finds that experts could prove 78% of them, suggesting most statements are mathematically sound, though the remaining 22% may include false or underspecified statements rather than just hard ones. The paper evaluates 7 prover models on the generated statements and releases the MathlibLemma benchmark for evaluating formal reasoning models.

**Compliance With Llm Reviewing Policy:**

Affirmed.

**Final Justification:**

The paper addresses a real problem (missing folklore lemmas in Mathlib) and the pipeline produces concrete results, including three Mathlib merges. The rebuttal answered my open questions: cost is about 700 dollars, the merge details clarify the gap between verified and mergeable, and the audit breakdown (25/31 missing hypotheses, 5/31 likely false) is informative. The benchmark entanglement concern is partially mitigated by Goedel outperforming GPT on the foundational subset. The novelty filter limitation (syntactic only, may miss semantic duplicates) remains acknowledged but unresolved. Overall the rebuttal reinforced my prior assessment. The contribution is solid and practically useful, though the scope of validation (3 merges, no downstream proving ablation) limits the strength of the claims.

**Key Questions For Authors:**

- The paper contributes both a pipeline and a benchmark. How do you see these two contributions relating to each other long-term? If the pipeline is replaced by better models, does the benchmark remain useful on its own, and if so, should it be constructed independently of any particular generation system?

- What was the approx. total API cost for producing the 4,028 type-checked statements? Table 1 shows ~9,148 candidates were proposed from 109 seed files, but the dollar cost of the full pipeline is not reported.

- How many proved lemmas were actually submitted for merging, and were any rejected? If so, what were the main reasons: correctness, proof style/verbosity, naming conventions, or redundancy? This would help clarify the gap between "verified" and "Mathlib-mergeable".

- Have you tried using different models for the Discovery agent? The current benchmark may be biased toward GPT-5.1's generation patterns, and a multi-model discovery pipeline might produce a more diverse benchmark.

- The missing-hypotheses failure mode (68%) is interesting. Have you tried prompting models to explicitly list all necessary conditions before generating the formal statement?

- The human audit samples 138 out of 2,216 unsolved instances using 2 lemmas per seed. How was the sampling stratified, and could you report confidence intervals on the 78% provability rate? Also, the audit only covers unsolved instances. Do you have any quality assessment for the 1,812 solved ones?

- The Judge rejects 46% of valid statements (false negatives). How does this affect benchmark coverage? Are there entire categories of folklore lemmas that the Judge systematically filters out?

- For the 22% of audited instances that experts could not prove, can you characterise the failures more fully? The deep-dive covers 31 instances with missing hypotheses as the dominant mode, but what accounts for the rest? Are some genuinely false, or just too hard?

**Limitations:**

The authors discuss deduplication limitations (syntactic only, may miss semantic duplicates), semantic drift risk in the repair loop, and the gap between verified and mergeable. These are honest and relevant. They do not discuss: (1) the cost of the pipeline, (2) the benchmark's dependence on the generation model, (3) the Judge's 46% false negative rate and its impact on benchmark coverage, or (4) the error analysis on the 22% of audited instances that experts could not prove is shallow (31-instance deep-dive, one dominant category reported) and does not fully characterise whether these are false, underspecified, or simply too hard. No ethical concerns.

**Strengths And Weaknesses:**

**Strengths:**

The task framing (discovering missing library lemmas rather than solving known problems) is a genuinely useful contribution, and the 3 Mathlib merges provide real validation. The human audit is well-done and the model complementarity finding is interesting. The missing inference cost analysis and the benchmark-pipeline entanglement are concerns, but the overall direction is valuable and the paper is thorough.

- The 3 Mathlib code merges are real external validation. The library has strict review standards, so this shows the pipeline can meet expert quality in some cases.

- The human audit is useful. Experts proved 107 out of 138 sampled unsolved instances (78%), suggesting most generated statements are mathematically sound. The remaining 22% may include false or underspecified statements, not just hard ones.

- The model diversity result is interesting. The union of all models solves more than 2x what the best individual model does, suggesting genuinely complementary capabilities across provers.

- Good reproducibility. Prompt templates, extraction logic, and hyperparameters are all documented.

**Weaknesses:**

- The paper reports 3 Mathlib merges but only submitted "a few representative lemmas", not all proved statements. Without knowing how many were submitted and how many were rejected, the merge rate is hard to interpret. There is also no cost analysis. Let's assume 600 dollar in total API costs, that is 200 dollar per merged lemma. Perhaps, a human formalizer could probably do the same in a couple of hours.

- The novelty filter may miss duplicates. It flags a statement as trivial only if `aesop` can prove it in a few steps. But a lemma could be a restatement of something already in Mathlib without `aesop` catching the connection (semantic duplicates). So some "new" lemmas may be redundant.

- The benchmark is entangled with the pipeline. GPT-5.1 discovers the lemmas, judges them, and formalizes them, so the benchmark reflects GPT-5.1's view of what folklore lemmas look like and how to phrase them. A model with a similar reasoning style might perform better not because it is a stronger prover, but because the problems were generated in a familiar style. The type-checking and proof verification are objective (Lean kernel), so evaluation is fair, but the selection of what gets evaluated is shaped by one model.

---

> ### Author Rebuttal · Authors · 2026-03-31
>
> Thank you for the positive assessment. Our main contribution is to formulate AI-assisted folklore-lemma mining from an existing formal library as a new task and to show that it is already feasible in practice. We will clarify this framing in the revision.
>
> > W1.1 / Q2 / L1. Approximate total API cost
>
> Total API cost was about \\$700: about \\$200 for generation/judge/formalization (producing the 4,028 type-checked statements in Table 1) and about \\$500 for downstream proving experiments. See also our response to Reviewer ns2T, L2 for the full cost breakdown and discussion of cost-effectiveness.
>
> > W1.2 / Q3. Mathlib submissions and rejection reasons
>
> We submitted 6 proved lemmas to Mathlib: 3 were merged, 1 was rejected as redundant, and 2 were not pursued further within the paper timeline. Thus, in this sample, the gap between “verified” and “Mathlib-mergeable” seems to come mainly from library curation criteria such as non-redundancy, suitable generality, naming/API fit, and maintainer bandwidth, rather than theorem validity. We will clarify this distinction in the revision. See also our response to Reviewer ns2T, W1 / Q2 for why we intentionally did not attempt bulk submission.
>
> > W2. Novelty filter / semantic duplicates
>
> We agree that the current novelty filter is incomplete, which is acknowledged in Section 7 (line 420-423), and we will make it more explicit in the revision. That said, we view this as a limitation of the filtering stage rather than a contradiction of the main feasibility claim. Some generated lemmas were turned into Mathlib PRs and merged, suggesting that the pipeline can surface genuinely useful results rather than only redundant restatements. Stronger novelty checking is an important direction for future work.
>
> > W3 / Q1 / Q4 / L2. Benchmark entanglement and alternative discovery models
>
> We believe the benchmark remains useful on its own even if future work replaces the current pipeline with stronger models. Even after excluding the first two rows as potentially entangled with GPT-based generation, the remaining results still show meaningful patterns, including the specialist-vs-generalist tradeoff and complementarity across provers; moreover, GPT does not dominate even in this potentially GPT-favoring setup, since Goedel still performs best on the foundational subset. We also ran preliminary discovery experiments with other models, including Qwen, but obtained substantially weaker candidates and no useful conclusions, so we did not pursue a systematic multi-model study.
>
> > Q5. Missing-hypotheses prompting
>
> Yes. We ran a small pilot with a revised discovery prompt that explicitly asks the model to check necessary conditions before producing the formal lemma. In this pilot, 123/160 generated statements compiled; after manually reviewing all 123 compilable statements, 117 were provable as stated, and only 2 of the 6 residual failures appeared to be genuine missing-hypothesis cases. In the corresponding old-prompt run, 123/160 also compiled, and only 2 of the 7 unresolved compilable cases we reviewed appeared to be genuine missing-hypothesis cases. So this modification is reasonable, but in our pilot, among compilable statements, missing hypotheses were not the dominant residual issue.
>
> > Q6. Audit sampling and solved-instance quality
>
> The audit sampled 2 unsolved lemmas per seed file, not a uniform random sample from all 2,216 unsolved instances, so we will clarify this in the revision and avoid presenting the 78% figure as a precise global estimate.
>
> For the 1,812 solved instances, the main quality signal is that they already have Lean-verified proofs. We also performed a small supplementary expert audit on 5 held-out seed files, examining up to 20 solved lemmas per seed. Experts proved 79 of 94 lemmas (84.0%), with mean per-seed rate 83.7% and sample standard deviation 4.6%. We will add this result and clarify its limited scope in the revision.
>
> > Q7 / L3. Judge false negatives and coverage
>
> We agree that a 46% false-negative rate reduces recall. However, it did not create seed-level coverage holes in our data: every seed still contributed surviving items, rather than some seeds being filtered out entirely. Thus, the Judge appears to reduce coverage within seeds rather than remove whole seed-defined topic areas. We did not perform a finer-grained category-level analysis, so we cannot rule out subtler topical biases, and we will clarify this limitation in the revision.
>
> > Q8 / L4. Characterization of the expert-unproved audited instances
>
> We revisited the 31 audited instances that experts left unproved. Missing hypotheses / underspecification remains the dominant failure mode (25/31). Of the remainder, 5/31 are likely false as stated (typically due to degenerate defaults or incompatible typeclass conditions), and only 1/31 appears to be a genuinely hard technical formalization gap with a plausibly true underlying statement. We will add this breakdown to the revision.

---

> > ### Author Rebuttal · Reviewer_rjii · 2026-04-04
> >
> > The cost breakdown is now clear (about 700 dollars total: 200 for generation/filtering/formalization, 500 for proving experiments). The merge details clarify the gap between "verified" and "mergeable" (6 submitted, 3 merged, 1 rejected as redundant, 2 not pursued due to timeline). The audit characterisation of the 31 expert-unproved instances is useful (25 missing hypotheses, 5 likely false, 1 genuinely hard). The benchmark entanglement concern is partially mitigated by the finding that Goedel outperforms GPT on the foundational subset despite GPT-5.1 generating the benchmark.

---

### Decision · Program_Chairs · 2026-04-30

**Decision:**

Accept (regular)

**Comment:**

The reviewers agree that this paper addresses a real and practically relevant problem, and they view the benchmark and verified lemma library as meaningful contributions. I also found the rebuttal helpful: it clarified the cost, the Mathlib submission outcomes, and several limitations around maintenance and evaluation, and I have taken those clarifications into account in my decision.

The main remaining weakness is that the paper does not directly validate its strongest practical motivation through a downstream usefulness experiment, and some of the novelty lies more in the task formulation than in the individual technical components. Overall, I find the work technically sound, well executed, and likely to be useful to part of the ICML community, so I recommend Weak Accept.